# Structural basis for antibody recognition of vulnerable epitopes on Nipah virus F protein

Patrick O. Byrne [1], Brian E. Fisher[2], David R. Ambrozak[3], Elizabeth G. Blade[1], Yaroslav Tsybovsky [4], Barney S. Graham [2,5], Jason S. McLellan [1] ✉ & Rebecca J. Loomis[2,6] ✉

Nipah virus (NiV) is a pathogenic paramyxovirus that causes fatal encephalitis in humans. Two envelope glycoproteins, the attachment protein (G/RBP) and fusion protein (F), facilitate entry into host cells. Due to its vital role, NiV F presents an attractive target for developing vaccines and therapeutics. Several neutralization-sensitive epitopes on the NiV F apex have been described, however the antigenicity of most of the F protein's surface remains uncharacterized. Here, we immunize mice with prefusion-stabilized NiV F and isolate ten monoclonal antibodies that neutralize pseudotyped virus. Cryo-electron microscopy reveals eight neutralization-sensitive epitopes on NiV F, four of which have not previously been described. Novel sites span the lateral and basal faces of NiV F, expanding the known library of vulnerable epitopes. Seven of ten antibodies bind the Hendra virus (HeV) F protein. Multiple sequence alignment suggests that some of these newly identified neutralizing antibodies may also bind F proteins across the *Henipavirus* genus. This work identifies new epitopes as targets for therapeutics, provides a molecular basis for NiV neutralization, and lays a foundation for development of new cross-reactive antibodies targeting Henipavirus F proteins.

Nipah virus (NiV) is an enveloped, negative-sense, single-stranded RNA virus in the *Henipavirus* genus of the *Paramyxoviridae* family[1]. NiV was first identified following an outbreak of severe encephalitis in humans who had been exposed to pigs in Singapore and peninsular Malaysia between 1998 and 1999[1–3]. This outbreak was associated with 265 cases of encephalitis, including 105 deaths, and the culling of approximately one million domesticated pigs[1]. Since the late 1990s, NiV has circulated in almost annual cycles in Bangladesh and India, including recent outbreaks in Kerala, India (2018, 2021). The Bangladesh strain of NiV has demonstrated human-to-human transmission and high mortality rates (60–70%)[4–15]. NiV outbreaks often begin with zoonotic exposure to the natural reservoir (fruit bats of the *Pteropodidae* family), infected

intermediate hosts such as pigs, cows, goats, horses, dogs, and cats[1,8,16–25] or indirect contact to food contaminated with animal secretions[11,26]. Human-to-human transmission can occur either through direct contact with bodily fluid or exposure to respiratory droplets[10,27–29].

The broad species tropism, high mortality rate, and human-to-human respiratory transmission highlights NiV's potential to endanger public health on a large scale. The World Health Organization, US Centers for Disease Control and Prevention, and the Coalition of Epidemic Preparedness Innovations have identified NiV, a BSL-4 pathogen, as a high-priority pathogen and pandemic threat[30]. A NiV vaccine antigen that could elicit a protective immune response, or antibodies

[1]Department of Molecular Biosciences, The University of Texas at Austin, 78712 Austin, TX, USA. [2]Viral Pathogenesis Laboratory, Vaccine Research Center, National Institute of Allergy and Infectious Diseases, National Institutes of Health, 20892 Bethesda, MD, USA. [3]Immunology Laboratory, Vaccine Research Center, National Institute of Allergy and Infectious Diseases, National Institutes of Health, 20892 Bethesda, MD, USA. [4]Vaccine Research Center Electron Microscopy Unit, Cancer Research Technology Program, Leidos Biomedical Research, Inc., Frederick National Laboratory for Cancer Research, 21701 Frederick, MD, USA. [5]Present address: Morehouse School of Medicine, 30310 Atlanta, GA, USA. [6]Present address: GSK Global Health R&D Vaccines (GVGH), 53100 Siena, Italy. ✉e-mail: jmclellan@austin.utexas.edu; rebeccajo.x.loomis@gsk.com

capable of neutralizing viral infection, would serve as expedient and critical medical countermeasures in the event of an outbreak.

NiV has two membrane-anchored glycoproteins, the attachment protein (G, also known as RBP (receptor binding protein)) and fusion (F) protein, that mediate receptor binding and entry into host cells. NiV F initially exists as a homo-trimer on the viral surface, where it adopts a prefusion conformation. Host proteases process the F protein into disulfide-linked subunits, F1 and F2. G/RBP binding to the cellular receptor, ephrin-B2 or -B3, triggers a conformational change in F that facilitates fusion of the viral and cellular membranes as F adopts the highly stable postfusion conformation[31–36]. Both the F and G/RBP proteins thus present attractive targets for vaccine design, and antibodies generated against either F or G/RBP have been shown to neutralize the fusion machinery and inhibit NiV infection[35,37–41]. Immunization of mice with the prefusion F protein elicits higher titers of neutralizing antibodies compared to immunization with postfusion F, revealing the prefusion conformation as a primary target for the F-specific immune response[32,42]. Neutralizing antibodies known to bind the prefusion F protein recognize epitopes on the apical surface of prefusion NiV F, located within domain 3[37–39], whereas the antigenic properties and neutralization sensitivity of the lateral and basal surfaces, composed of domains 1 and 2, remain uncharacterized.

Here, we immunize mice with a rationally designed, prefusion-stabilized trimeric form of NiV F and identify ten antibodies that neutralize NiV in an in vitro infection assay. All ten of these antibodies bind to F in its prefusion conformation. We characterize the three-dimensional structures of all ten neutralizing antibodies and classify their modes of binding to eight epitopes. Neutralizing antibodies decorate most of the prefusion F protein's surface. Four antibodies recognize previously uncharacterized neutralization-sensitive epitopes on the lateral and basal surfaces of NiV F, located within domains 1 and 2. The remaining six antibodies recognize sites that partially overlap with known epitopes in domain 3. Sequence alignments with multiple Henipavirus fusion proteins show that the identified neutralizing antibodies bind to highly conserved pockets of the protein suggesting potential to cross-react with other Henipaviruses. This work identifies new sites of vulnerability on NiV F and provides a basis for developing vaccines and immunotherapeutics targeting an array of vulnerable epitopes.

## Results

### Immunization of mice with NiV F and isolation of monoclonal antibodies

CB6F1/J mice were immunized with NiV F protein stabilized in its prefusion conformation (previously referred to as NiVop08, derived from the Malaysian strain) as described in the Methods Section (Fig. 1A, Supplementary Table 1)[42]. The mice were boosted at week 34 and 42 following an initial immunization series (week 0, 3 and 10). The time between the initial immunization series and the boosts used for B cell sorting allowed for maturation and increased affinity of the antibody response. After assessing mouse sera for antigen binding and virus neutralization capability, isolated splenocytes were screened against an established antibody panel to identify B cells specific to the included NiV prefusion F-conjugated probe, which were single-cell index-sorted by fluorescence-activated cell sorting (FACS) into 96-well plates (Supplementary Figure 1A-D, F,G). The resulting sorted B cells were sequenced to obtain paired heavy and light chain sequences. Of the 384 single cells collected and analyzed from prefusion NiV F probe-positive B cells, 96 paired heavy and light chain sequences were identified (Fig. 1B, Supplementary Figure 1A-D, F, G).

Fifty-three monoclonal antibodies, largely from different VDJ lineages, were selected for further analysis. Variable heavy and light chain sequences were cloned into human IgG or light chain kappa expression plasmids and were expressed, purified, and tested for binding specificity. Of the antibodies tested, 21 showed a strong binding preference for prefusion NiV F, whereas 14 antibodies bound to both prefusion and postfusion NiV F (Supplementary Fig. 1E) and 18 did not bind to either prefusion or postfusion NiV F. Negative stain electron microscopy confirmed that a subset of these antibodies bound to prefusion NiV F (Fig. 1C). We then tested the ability of NiV F antibodies to neutralize the pseudotyped virus. The assay used a variant of vesicular stomatitis virus (VSV) that lacks the native VSV G glycoprotein (VSVΔG) but includes luciferase, NiV F and NiV G/RBP (hereafter, "NiV F/G VSVΔG-luciferase"). Serial dilutions of each monoclonal antibody were used to assess their neutralization capability, similar to what has been described previously[43,44]. Ten of the 53 monoclonal antibodies displayed neutralizing activity and of those, nine bound exclusively to the prefusion conformation and one bound to both pre- and postfusion conformations (Fig. 1D, Supplementary Figure 1E, Supplementary Table 2). Previously characterized monoclonal antibodies 5B3 and 066 were used as positive controls[37,38]. Lastly, we performed a competition binding experiment using biolayer interferometry (BLI). Prefusion NiV F was adhered to a biosensor tip, which was then sequentially dipped into solutions of competitor and analyte antibody. This series of competition BLI experiments (Fig. 1E, Supplementary Figure 1H) yielded distinct but overlapping binding fingerprints for each of the isolated antibodies. Three antibodies (1F3, 4H3 and 2D3) competed with 066, which binds to the NiV F apex[37]. Five antibodies (4H3, 2D3, 1H8, 4B8, and 1F2) competed with 5B3, which binds between the lateral and apical surfaces of NiV F[38]. Four antibodies (2B12, 4F6, 1H1 and 1A9) did not compete with either control antibody, suggesting that they might bind to the lateral or basal sides of NiV F[37,38].

### Cryo-EM structures of prefusion NiV F complexed with ten neutralizing antibodies

To examine the binding properties of neutralizing NiV F antibodies, we performed surface plasmon resonance (SPR) (Fig. 2). Six antibodies were chosen based on their neutralizing activity and their binding fingerprint: 4H3, 2D3, 1H8, 1A9, 1H1 and 2B12. All six antibody fragments of antigen binding (Fabs) bound to prefusion-stabilized NiV F with high affinity, with $K_D$ values in the range of 1-25 nM for Fabs 4H3, 2D3, 1A9, 1H1 and 2B12 (Fig. 2A-F, Table 1). Antibody 1H8 exhibited potent binding affinity with a slow dissociation rate near the detection limit of the instrument, requiring us to estimate an upper bound for the $K_D$ at around 20 pM. Fab 2B12 exhibited distinct binding kinetics consisting of a fast association rate and a fast dissociation rate. We also confirmed the binding of the remaining four neutralizing antibodies by BLI, with $K_D$ values of 37 nM and 9 nM for 1F3 and 4B8, respectively (Supplementary Figure 2A-D). Antibodies 1F2 and 4F6 exhibited slow off-rates, requiring us to set an upper limit of ~1 nM for their $K_D$ values. We observed no correlation between Fab binding affinity and neutralization potency (Table 1, Supplementary Figure 2E).

To better understand the interactions between these neutralizing antibodies and prefusion NiV F, we performed structural studies by cryo-EM. Initial screening of Fabs complexed with NiV F resulted in aggregated particles, which prevented further analysis. The addition of amphipol A8-35 to a final concentration of 0.1% (w/v), just prior to incubation with Fab, reduced aggregation and allowed us to obtain single particle cryo-EM reconstructions at high resolution (≤ 3.0 Å) for six complexes (Fig. 3, Supplementary Figures 3-9, Supplementary Table 3). Distinct stoichiometric classes were separated during 3D refinement. 3D reconstructions for four of the antibody complexes (4H3, 1H1, 1A9 and 2B12) exhibited orientation bias during screening, which was overcome by collection at 30° tilt.

Each of the six antibodies bound to a distinct epitope, with the various epitopes scattered across the surface of the prefusion NiV F protein (Figs. 3, 4). Antibody 4H3 binds an apical epitope within domain 3, burying 703 Å² of surface area on NiV F. The interface includes one π−π stacking interaction and four hydrogen bonds. The

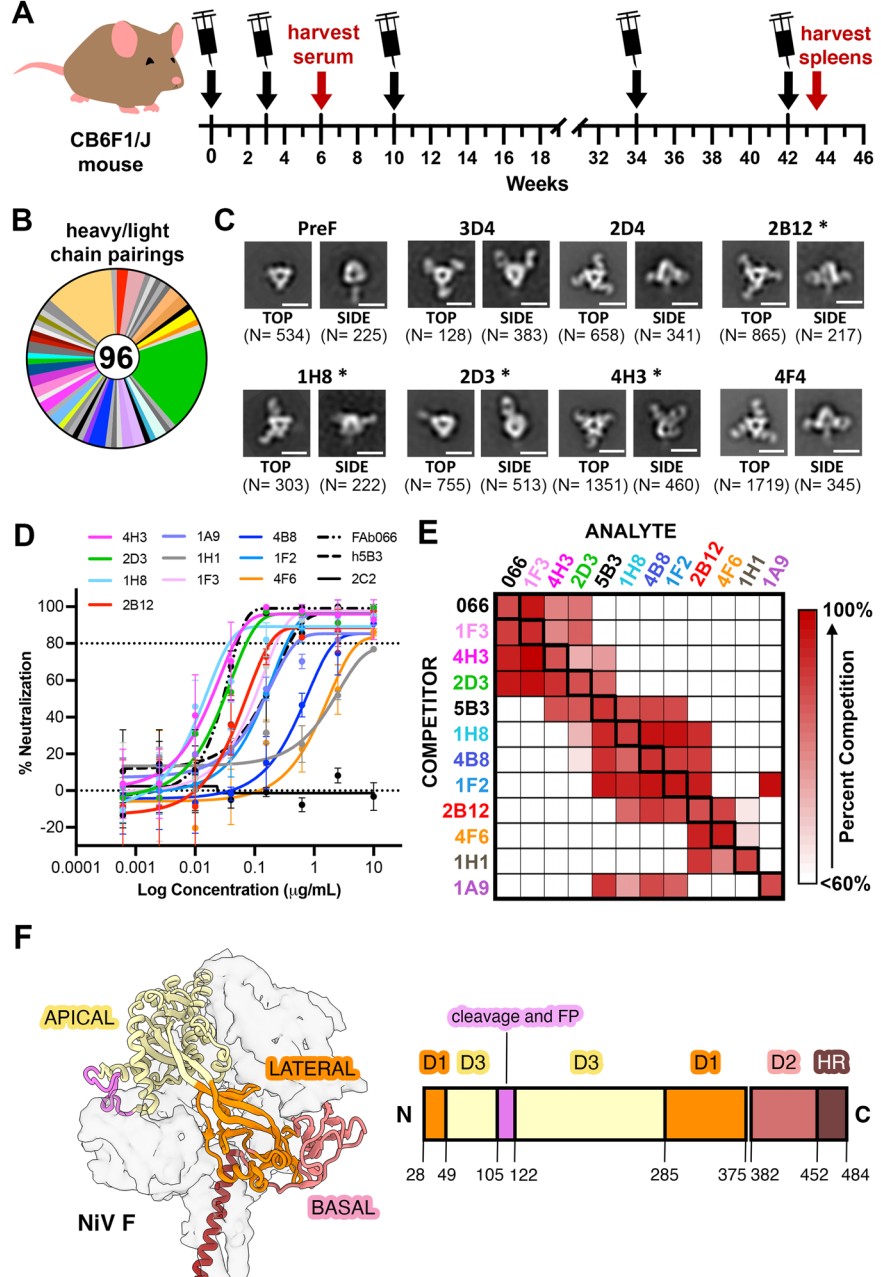

**Fig. 1 | Isolation and characterization of antibodies that bind to NiV F.**
**A** Schematic of immunization of CB6F1/J mice. Black arrows with syringes denote times of immunization, the red arrows denote the times of serum harvest or spleen harvest. **B** Pie chart representing mouse paired heavy-light chain sequences recovered from B cells of mice immunized with prefusion NiV F. FACS gating/ sorting strategies are shown in Supplementary Fig. 1F, G and described in the Methods Section. **C** Representative 2D class averages from negative stain EM of prefusion NiV F, alone and complexed with Fabs. Numbers of particles in each class are listed in parentheses below each image. Details regarding the numbers of micrographs are included in the Methods Section. Numbers of particles in each class are shown in parentheses below each image. Asterisks indicate antibodies that were shown to be neutralizing in the pseudovirus neutralization assay (Fig. 1D). **D** Neutralization of pseudotyped virus. Error bars show the standard deviation of the mean for three technical replicates. Data is representative of two independent

experiments. 2C2 is a non-neutralizing antibody. **E** Competition binding experiment, as measured by biolayer interferometry. Competitor antibodies are listed on the left by row, analyte antibodies are listed on the top by column. The percent competition is indicated by intensity of shading, from white (<60%) to red (100%). A value less than 60% is considered to not compete. Thicker boxes along the diagonal indicate self-antibody competition controls. Larger competition binding matrix is shown in Supplementary Figure 1H, highlighted here is the neutralizing antibodies competing with one another. Competition binding among neutralizing antibodies is representative of two independent experiments. **F** Diagram of the structure of prefusion NiV F. One protomer is shown as a ribbon diagram, and the remaining two protomors are shown as simulated gray traces at ~5 Å resolution. A primary sequence diagram is shown at the right. Domain 1 (D1/lateral) is colored orange, domain 2 (D2/basal) is colored red and domain 3 (D3/apical) is colored yellow. The heptad repeat is colored brown.

epitope abuts the N-acetylglucosamine at Asn67, however, the map quality was insufficient in this region to model an interaction between 4H3 and the glycan itself (Fig. 3A, 4A). Antibody 2D3 binds a quarternary apical epitope on domain 3, comprising four hydrogen bonds

and 699 Å² of buried surface area on NiV F (658 Å² on one protomer and 41 Å² on the other) (Figs. 3B, 4B). Antibody 1H8 binds to a quaternary epitope spanning domains 1 and 3 on two adjacent protomers. The interface between NiV F and Fab 1H8 buries 1132 Å² of surface area

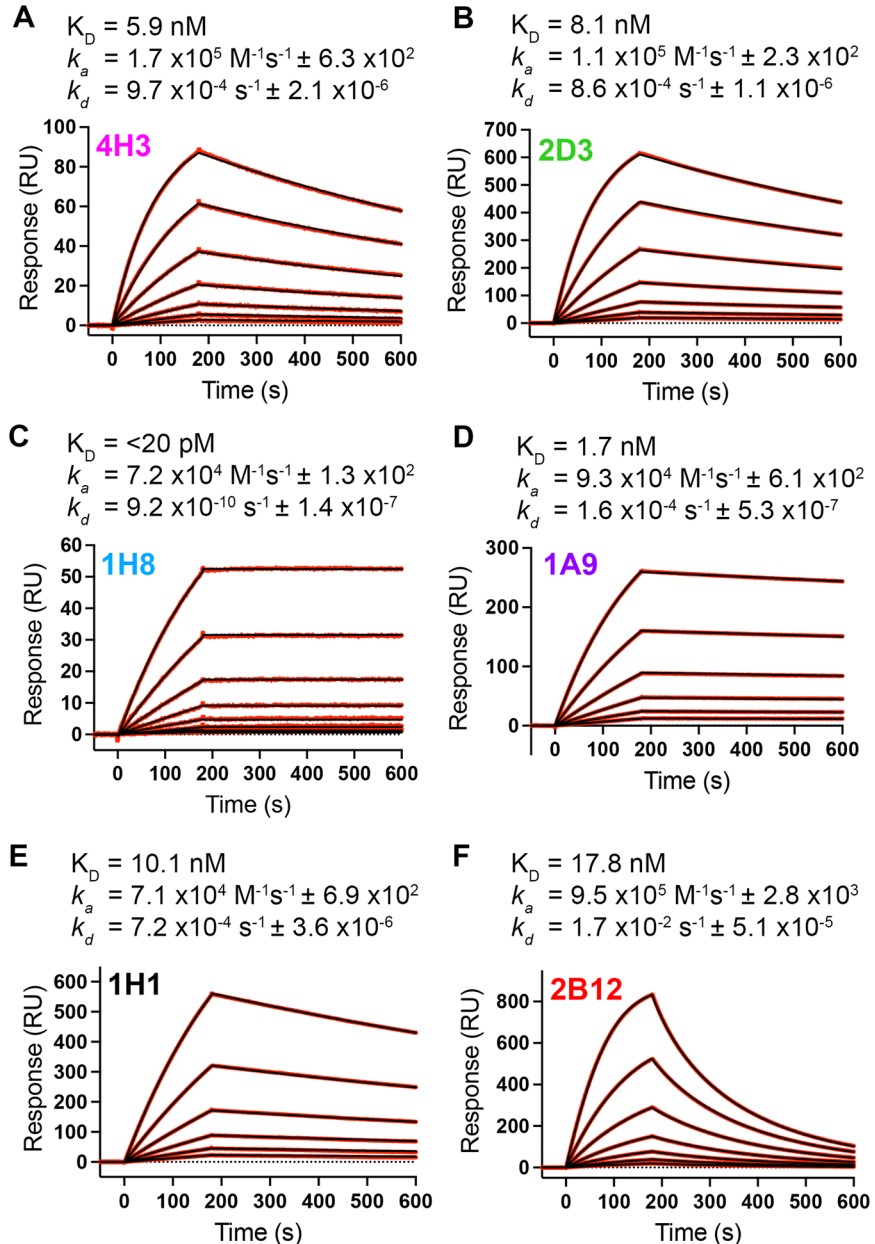

**Fig. 2 | Neutralizing antibodies bind to prefusion NiV F with high affinity.**
**A**–**F** Binding of Fabs from six neutralizing antibodies to NiV F, measured by surface plasmon resonance. The equilibrium dissociation constant ($K_D$), the association ($k_a$) and dissociation ($k_d$) rate constants are indicated above each graph. Experimental binding curves are shown as red traces, with the best fit values to a 1:1 binding model shown as black lines. Data are representative of two biological replicates and two technical replicates.

(984 Å$^2$ on one protomer, 148 Å$^2$ on the other), featuring six hydrogen bonds and two π–π stacking interactions: one between the heavy chain and protomer 1, the other between the light chain and protomer 2 (Fig. 3C, 4C). Antibody 1A9 binds to an interface encompassing a large quaternary epitope spanning domains 1 and 2 on two adjacent protomers, burying 1302 Å$^2$ on NiV F (1025 Å$^2$ on one protomer, 277 Å$^2$ on the other; Figs. 3D, 4D). The 1A9 interface comprises a variety of interactions, including six π–π stacking interactions and twelve hydrogen bonds. Notably, the framework region of the 1A9 heavy chain forms a hydrogen bond and van der Waals contacts with a surface-exposed portion the fusion peptide (Fig. 4D). Antibody 1H1 binds to a quaternary epitope spanning domains 2 and 3 on adjacent protomers. The 1H1 interface includes nine hydrogen bonds and buries 725 Å$^2$ of surface area on NiV F (633 Å$^2$ on one protomer, 92 Å$^2$ on the other), including a buried charged residue (Figs. 3E, 4E). Finally,

antibody 2B12 recognizes an epitope within domain 2 on the membrane-proximal surface of NiV F, adjacent to the epitope recognized by antibody 1H1. The 2B12 interface comprises 11 hydrogen bonds and 892 Å$^2$ of buried surface area (855 Å$^2$ on one protomer, 37 Å$^2$ on the other protomer; Fig. 3F, 4F). Our prefusion-stabilized NiV F variant thus elicits antibodies against the apical, lateral, and basal surfaces.

We also determined the epitopes for the four other neutralizing antibodies by cryo-EM to modest resolution: Fab 1F3 to 4.3 Å, Fab 1F2 to 5.3 Å, Fab 4B8 to 5.8 Å and Fab 4F6 to 4.1 Å (Supplementary Figure 10). These resolutions were sufficient to conclude that the antibody epitopes overlapped with those of the other antibodies reported in this paper. Antibody 1F3 binds to the apex of the NiV F protein on domain 3, overlapping partially with the epitope of antibody 4H3 (Supplementary Figure 10A). Antibodies 1F2 and 4B8 bind to the same

epitope as antibody 1H8, spanning domains 1 and 3 (Supplementary Figure 10B-C). The CDR sequences of 1H8, 4B8, and 1F2 exhibited high sequence similarity, consistent with their shared binding epitope (Supplementary Figure 10D). Antibody 4F6 binds to domain 2, exhibiting partial overlap with antibody 2B12 while also extending into a novel epitope on the membrane-proximal face of NiV F (Supplementary Figure 10E).

Our prefusion-stabilized variant elicits a variety of antibodies that decorate most of the NiV prefusion F protein surface. Viewed together, this group of neutralization-sensitive sites exhibits two interesting features. First, the epitopes cluster in regions that avoid the four N-linked glycosylation sites (Asn67, Asn99, Asn414, Asn464). Indeed, modeling of the branched glycans reveals that NiV F bears three glycan straps, each one extending from the base to the apex (Fig. 5A). Second, a multiple sequence alignment of F proteins within the *Henipavirus* genus indicates that neutralizing antibodies reported here recognize pockets of conserved surface residues (Fig. 5B-D, Supplementary Figure 11), suggesting that neutralizing antibodies elicited by NiV F might also recognize related Henipavirus F proteins.

To investigate the breadth of antibody recognition across viruses, we compared the binding of neutralizing antibodies to both NiV F and Hendra virus (HeV) F. We chose a subset of antibodies from our study that bind the apical, lateral, or basal face of NiV F and assessed binding to HeV F by BLI (Supplementary Figure 12). All antibodies tested bound both NiV F and HeV F. Antibodies directed to the apical and basal faces exhibited no differences in binding to NiV F and HeV F (2D3, 1H8, 1H1, 2B12, 4F6), however, two antibodies directed to the lateral face of NiV F (1A9 and 1F2) displayed reduced binding to HeV F when compared to NiV F. Indeed, sequence conservation is greatest in the apical and basal surfaces (Fig. 5B-D), suggesting that this region may better tolerate antigenic drift. On the contrary, antibody 1H8 binds the lateral face but shows equal preference for NiV F and HeV F. Antibodies that recognize the basal face (1H1 and 2B12) exhibit especially high sequence conservation (Supplementary Figure 11).

We attempted to produce a prefusion-stabilized F protein from Cedar virus (CedV), which has lower sequence conservation with NiV F, but we were unsuccessful. We transferred the prefusion-stabilizing substitutions from NiV F to CedV F, however, the protein expressed poorly and showed low stability[42]. This may be due to the low overall sequence conservation between NiV F and CedV F. Indeed, the residues surrounding the sites of disulfide and cavity-filling substitutions in NiV F (L172F, L104C/I114C) align especially poorly with CedV F and Ghanaian bat virus (GhV) F (Supplementary Figure 11). Further work will be required to engineer prefusion-stabilized forms of the Mòjiāng virus, CedV, and GhV F proteins, construct pseudotyped virus neutralization assays, and experimentally test the breadth of antibody binding and neutralization.

## Discussion

Neutralizing antibodies to NiV bind to two viral surface glycoproteins, G/RBP and F[35,37,39,40,45,46]. Recently isolated and characterized F monoclonal antibodies are prefusion-specific and directed to epitopes near the apex of NiV F located within domain 3[37–39]. We identified ten neutralizing antibodies directed against NiV F, all of which proved amenable to structural characterization. Six of these ten antibodies (4H3, 2D3, 1H8, 1F2, 1F3, 4B8) bind to the NiV apex in domain 3, consistent with previous reports[37,39]. While each of the domain 3-directed antibodies reported here exhibit partial overlap with known NiV F antibodies, they nevertheless employ unique modes of recognition by engaging additional surface residues outside previously defined epitopes. The apical domain 3 of NiV F therefore contains a variety of vulnerable antigenic sites.

The remaining four neutralizing antibodies elicited by prefusion-stabilized NiV F (1H1, 1A9, 2B12, 4F6) bind to an array of epitopes located within domains 1 and 2, on the lateral and basal surfaces of NiV

**Table 1 | Antibody Neutralization and Binding Data**

| Antibody | IC$_{80}$ (ng/mL) | R$^2$ (IC$_{80}$) | K$_D$ ± SD (nM) |
|---|---|---|---|
| 4H3 | 41.9 | 0.89 | 7.5 ± 2.0 |
| 2D3 | 74.9 | 0.88 | 11.1 ± 2.6 |
| 1H8 | 54.0 | 0.91 | <0.02 |
| 1A9 | 2.1 | 0.93 | 2.3 ± 1.2 |
| 1H1 | ND | 0.83 | 15.0 ± 4.7 |
| 2B12 | 121.0 | 0.89 | 22.6 ± 7.4 |
| 1 F3 | 253.3 | 0.94 | 37.3 ± 17.2 |
| 4B8 | 2.3 | 0.91 | 9.3 ± 6.1 |
| 1F2 | 459.0 | 0.98 | <1 |
| 4F6 | ND | 0.72 | <1 |
| h5B3 | 626.6 | 0.87 | 10* |
| 066 | 55.0 | 0.97 | ND |
| 2C2 | ND | ND | ND |

*Dang et al. (ref. [39]).

F. Importantly, these novel domain 1 and 2 epitopes include residues that are strongly conserved among known *Henipavirus* F protein sequences. Two of these novel epitopes reside solely within domain 2 (2B12 and 4F6). Another novel epitope, bound by antibody 1H1, comprises a quaternary interface that spans domains 2 and 3. The last novel epitope, bound by antibody 1A9, exhibits two interesting features; (i) it bridges a quaternary interface that spans domains 1 and 2 (ii) it includes the fusion peptide, the first such epitope discovered in the paramyxovirus family. The NiV F protein resembles other viral fusion proteins, such as human immunodeficiency virus 1 Env and influenza HA, with a known neutralization-sensitive epitope containing the fusion peptide[47,48]. Interestingly, the binding affinity of NiV F Fabs does not correlate with their neutralization potency, which suggests a complex relationship between antibody binding and F protein function during fusion. The ten antibodies described here expand our knowledge and understanding of neutralizing epitopes and antigenic surfaces on the prefusion NiV F protein. Each of these newly identified and characterized neutralizing antibodies harbor therapeutic potential.

Glycosylation of NiV F appears to play an important role in viral defense from the immune system. Three belts of N-linked glycans (12 total, 4 on each protomer) extend in opposing directions from the base to the apex of NiV F. Much of the lateral face of NiV F remains unshielded, which may contribute to the variety of vulnerable antigenic sites. Glycan shielding of envelope proteins plays a role in immune evasion in other viral class I fusion proteins, such as Ebola virus GP, influenza HA, Lassa virus GPC, HIV-1 Env and coronavirus S proteins (reviewed in[49]). NiV F harbors fewer glycosylation sites than other class I fusion proteins (4 N-linked sites in 546 residues of NiV F, compared to 22 N-linked sites in 1273 residues of the severe acute respiratory syndrome coronavirus 2 (SARS-CoV-2) spike protein), however, our results suggest that they provide sufficient steric hindrance to direct the immune response to surfaces with less glycan shielding[50]. Moreover, mutation of NiV F N-linked glycosylation sites is known to increase fusogenicity in an additive fashion[50], which suggests that glycosylation itself may stabilize the prefusion conformation.

Through multiple sequence alignment within the *Henipavirus* genus, the identified NiV-neutralizing antibody binding epitopes are in regions that have a high degree of conservation, particularly those that bind to the basal surfaces. NiV F antibodies also bind HeV F, however, the sequence conservation between NiV and HeV is high (~87% identity) and our multiple sequence alignments show that certain antibodies (e.g 4H3, 1H1 and 2B12) bind epitopes with greater sequence conservation than others. With our expanded knowledge of neutralizing epitopes and antigenic surfaces, the antibodies described

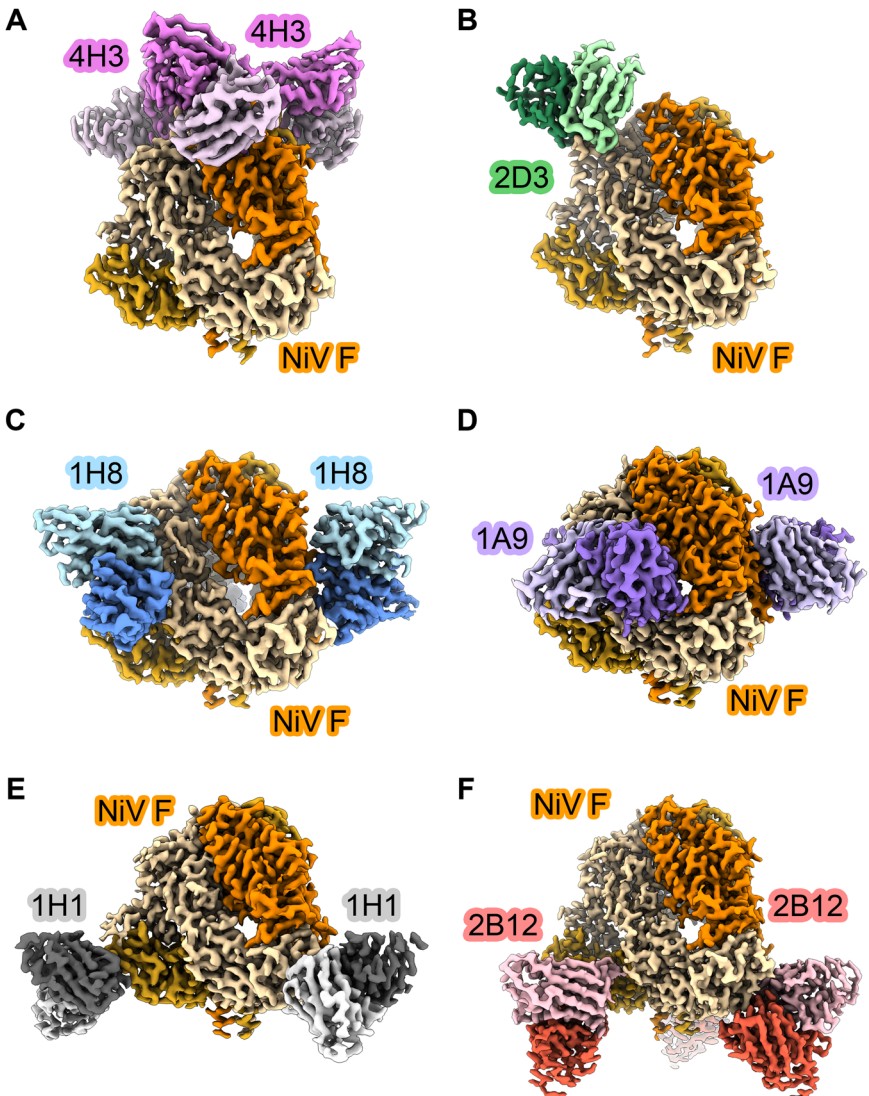

**Fig. 3 | Cryo-EM maps of prefusion NiV F in complex with neutralizing antibodies.** Sharpened 3D reconstructions showing side views of Fabs in complex with NiV F prefusion trimer. NiV F protomers are shown as shades of orange/yellow/brown. **A** Fab 4H3 is colored pink, **B** Fab 2D3 is colored green, **C** Fab 1H8 is colored blue, **D** Fab 1A9 is colored purple, **E** Fab 1H1 is colored black and **F** Fab 2B12 is colored red. Heavy chains for Fabs are darkly shaded, light chains are lightly shaded. The stoichiometry of the NiV F complex with Fab 2D3 was 1:1 (Fab:NiV F trimer), while the other complexes were all 3:1 (Fab:NiV F trimer).

here should serve as useful reagents for both the development of effective therapeutic antibodies and the study of the humoral immune response to natural *Henipavirus* infections.

Vaccines may play an important role in mitigating the spread and severity of NiV infection, yet no vaccine is currently available. Prefusion-stabilization of class I viral fusion proteins elicits higher neutralizing antibody responses than postfusion antigens, for example, respiratory syncytial virus (RSV) F, parainfluenza virus 1-4 F and SARS-CoV-2 spike[51–54]. The same trend holds for NiV: immunization with prefusion NiV F elicits high titers of neutralizing antibodies in mice, whereas postfusion NiV F elicits barely detectable neutralizing antibody titers[42,55]. Antibody 1H1 exhibits binding capacity for both prefusion and postfusion NiV F, suggesting there may be vulnerable epitopes preserved between both conformations. This phenomenon has been observed in the related pneumovirus RSV F protein, which contains neutralization-sensitive epitopes shared by both the prefusion and postfusion conformations[56,57].

NiV presents a clear threat to public health due to its zoonotic route of transmission from bats to humans and almost annual regional outbreaks characterized by human-to-human transmission and high fatality rates. This unmet medical need requires effective countermeasures to combat future pandemic threats. The National Institute of Allergy and Infectious Diseases has supported use of NiV as a prototype pathogen to identify generalizable approaches for better vaccine antigen design within the paramyxovirus family as part of its broad pandemic preparedness intiatives. The work described here identifies new sites of immunologic vulnerability on the NiV fusion protein and underscores the importance of prefusion stabilization in the design of paramyxovirus vaccine antigens.

## Methods

### Protein/Probe expression and purification

NiV prefusion F glycoprotein and avi-tagged probes and NiV postfusion F glycoprotein (derived from the Malaysian strain of Nipah virus) were expressed by transfection in 293 Freestyle (293 F) cells (Thermo Fisher Scientific, #12338026) using Turbo293 transfection reagent (SPEED BioSystem, #PX1002) according to the manufacturer's protocol. Sequences are in Supplementary Table 1. Transfected cells were incubated in shaker incubators at 120 rpm, 37 °C, 9% $CO_2$ overnight. On the second day, one tenth culture volume of

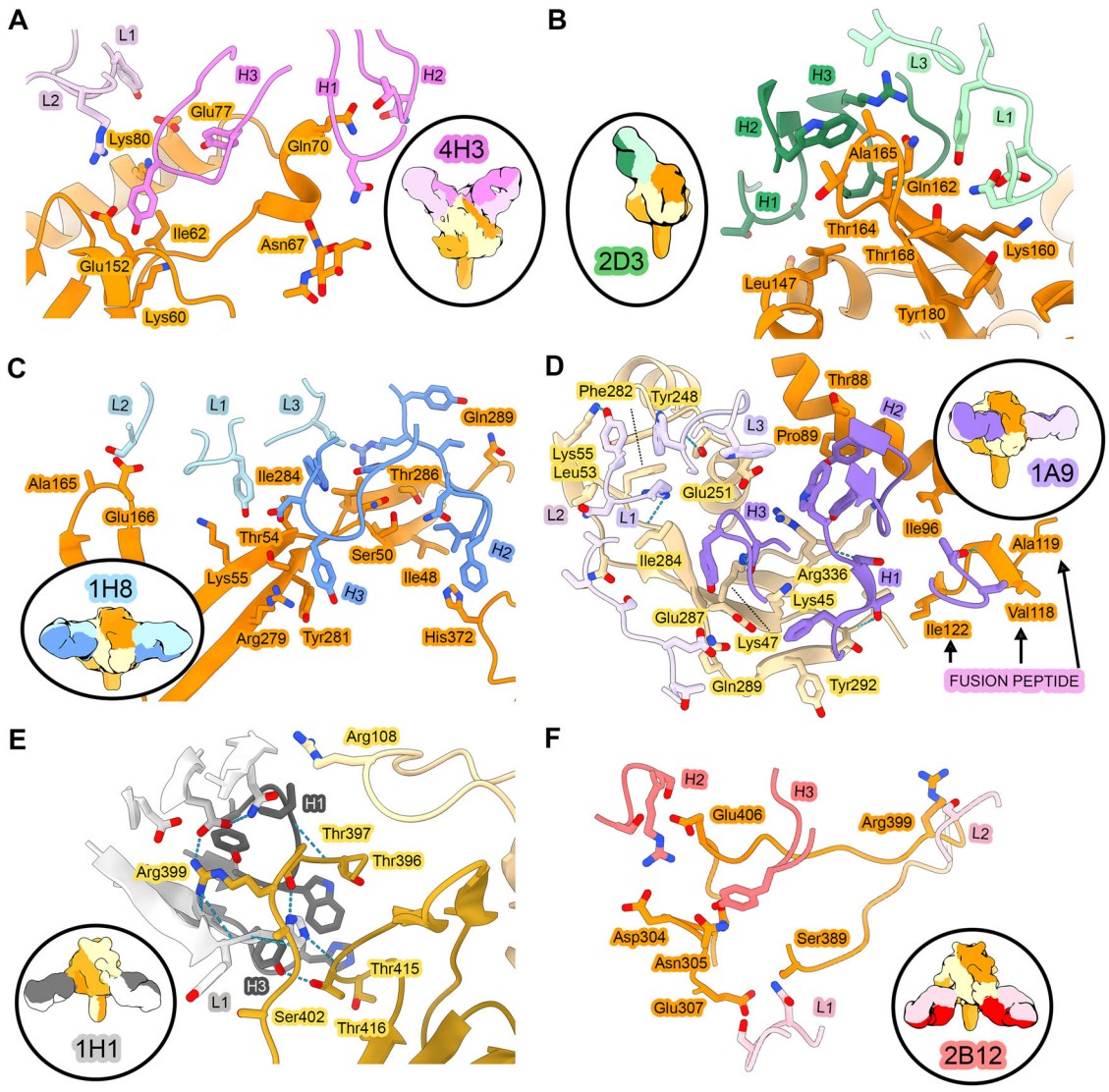

**Fig. 4 | Structures of neutralizing antibodies bound to prefusion NiV F.** Zoomed views of interfaces between Fabs and NiV F. Hydrogen bonds are indicated with light blue dashed lines. NiV F protomers are shown as shades of orange/yellow/brown. (**A**) Fab 4H3 is colored pink, (**B**) Fab 2D3 is colored green, (**C**) Fab 1H8 is colored blue, (**D**) Fab 1A9 is colored purple, (**E**) Fab 1H1 is colored black and (**F**) Fab 2B12 is colored red. Heavy chains for Fabs are darkly shaded, light chains are lightly shaded. Elliptical insets at the left or right of each panel show gaussian filtered 3D reconstructions of each overall complex, colored according to the same scheme.

Cell Booster medium (ABI Scientific, #PB2250) was added to each flask of transfected cells and cell cultures were incubated at 120 rpm, 37 °C, 9% $CO_2$ for an additional 4 days. Five days post-transfection, cell culture supernatants were harvested and proteins were purified from the supernatants using tandem $Ni^{2+}$ (Sigma Aldrich, #5893801001) and Strep-Tactin (IBA, #2-1201-025) affinity purification. The C-terminal purification tags of NiV F glycoprotein trimers used for immunization were removed by thrombin digestion at room temperature overnight and proteins were further purified by SEC in a Superdex 200 column (GE, #GE28-9909-44) in 1x phosphate-buffered saline (PBS). The tags on the probes were left intact and further purified by SEC in a Superdex 200 column in 1x PBS.

**Negative-stain electron microscopy**
Proteins were diluted to approximately 0.01-0.02 mg/mL with 10 mM HEPES, pH 7.0, 150 mM NaCl, adsorbed to a freshly glow-discharged carbon-coated grid, washed with the same buffer, and stained with 0.7% uranyl formate. Datasets were collected at a magnification of 100,000 using SerialEM on an FEI Tecnai T20 microscope equipped with a 2k x 2k Eagle CCD camera and operated at 200 kV[58]. The number

of micrographs in a given dataset ranged from 50 to 243 images. The nominal magnification was 100,000 and the pixel size was 0.22 nm. Particles were selected from micrographs automatically, followed by manual correction using EMAN2, when necessary[59]. The number of initial particles extracted for 2D classification ranged from 3055 to 37,357 particles. Reference-free 2D classifications were performed with Relion 1.4[60].

**Animal immunizations**
All animal experiments were reviewed and approved by the Animal Care and Use Committee of the Vaccine Research Center, NIAID, NIH (ACUC animal study protocol #VRC-17-709) and all animals were housed and cared for in accordance with local, state, federal and institute policies in an American Association for Accreditation of Laboratory Animal Care (AAALAC)-accredited facility at the NIH. Groups of ten 6-8 week old CB6F1/J female mice (Jackson Laboratory) were immunized at weeks 0, 3 and 10 weeks intramuscularly with 10 µg of recombinant NiV prefusion F antigens combined with 100 µg aluminum hydroxide (alum) in a volume of 100 µL (50 µL/leg). Serum was collected at week 6 and 12 following immunization. Week 6 sera were

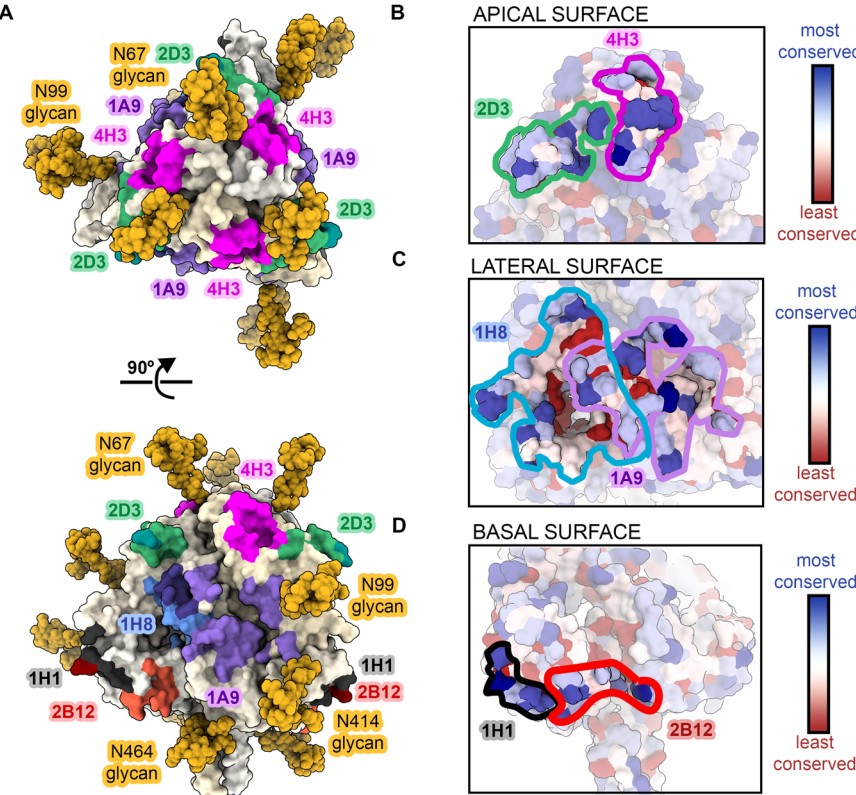

**Fig. 5 | Neutralizing antibodies avoid glycan straps to recognize conserved epitopes on prefusion NiV F.** Surface representation of prefusion NiV F. Individual NiV F protomers are colored in shades of pale yellow and glycans are colored in gold. Neutralizing antibody epitopes reported in this paper are painted according to the colors in Figs. 3–4. **A** Top and side views. **B**–**D** Zoomed views of the apical

(**B**), lateral (**C**) and basal (**D**) surfaces of NiV F. Surface residues on the F protein are colored by sequence conservation (red is least conserved; blue is most conserved). Sequence conservation is based on the alignment shown in Supplementary Figure 11. Solid lines denote the borders of antibody epitopes.

assessed for immunogenicity in enzyme-linked immunosorbent assays and for neutralization in VSVΔG-luciferase pseudovirus neutralization assay in vitro[42,43]. Prefusion F immunized mice were also immunized at week 34 and week 42 for B cell probe sorting. We boosted mice at week 34 to perform probe titration and B cell sorting in the desired 10-20 day peak response following boost. Our first sort did not yield the percentage of IgG-positive single cells we were expecting due to a generous IgG vs IgM gating strategy. We waited for the antibody response to start to wane, boosted again at week 42 and used a less generous IgG vs IgM gating strategy. The interval between the B cell sorting boosts (week 34 and 42) and the initial immunization series (week 0, 3 and 10) allow for maturation and increased affinity of the antibody response. Female mice were used in this study because male mice are more aggressive. Subsequent studies in ferrets showed no variability in immunogenicity between the sexes.

**Splenocyte isolation and staining for flow cytometry**
Mice were sacrificed with $CO_2$ and spleens harvested into 5 mL RPMI, 10% FBS, 1% Pen/Strep (1-2 spleens/tube) in gentleMACS tube (Miltenyi Biotec, #130-096-334). Spleens were mashed using SPLEEN_01 program on gentleMACS machine, strained through a 70 mm strainer and centrifuged at 850 x $g$ for 10 min at room temperature. The pellet was resuspended in 5 mL ACK Lysis Buffer (ThermoFisher, #A1049201)/ Spleen (2 spleens = 10 mL), lysed for 5 min at room temperature, equal volume of RPMI, 10% FBS, 1% Pen/Strep was added and centrifuged at 700 x $g$ for 8 min at room temperature. Pellet was resuspended in 10 mL 1x PBS, centrifuged at 500 x $g$ for 5 min at room temperature. The pellet was resuspended in 3 mL 1x PBS and transferred to 5 mL FACS tube, centrifuged, resuspended in 100 µL 1x PBS + .125 µL Aqua (Invitrogen, #L34957) or UV Blue (Invitrogen, #L23105) and incubated

15-20 min at room temperature, covered. Samples were washed with 1 mL 1x PBS and centrifuged. The pellet was resuspended in 50:1 $F_c$ Block (BD, #553142):BV Buffer (BD, #566349), incubated 2-5 min at room temperature, antibody mix (1:1 ratio with the $F_c$ Block + BV Buffer solution) was added and incubated at 4 °C for 20 minutes. Antigen-specific B cells were identified with a panel of ligands, including fluorescently labeled antibodies for B220, F4/80, Gr-1, CD4, CD8, IgG and IgM. Splenocytes were stained with an antibody cocktail consisting of anti-B220-BV421 (Biolegend, #103240), F4/80-BV510 (Biolegend, #123135), Gr-1-BV510 (Biolegend, #108438), CD4-BV510 (Biolegend, #100449), CD8-BV510 (Biolegend, #100752), IgG(1/2/3)-FITC (BD Pharmingen, #553443, #553399, #553403), and IgM-PE-Cy7 (Southern Biotech, #1140-17) with NiVop08 F-SA-APC. In addition, aqua blue was used to exclude dead cells. For titering of antibodies or probe; enough volume was added to cover all titration samples: 50 µL / sample and distributed to dilution tubes. Dilution tubes started with 20 µL antibody in 80 µL BV Buffer and were 8-point, 2x dilutions. Samples were pipetted up and down 15+ times, 50 µL was transferred to a new tube with 50 µL BV Buffer, pipet, transfer, repeat for the entire dilution series. Dilution antibody (probe) was incubated with antibody mix from above (50 µL + 50 µL) for 20 min at 4 °C. Samples were washed with 2 mL FACS Buffer (1x PBS, 0.1% sodium azide) and centrifuged. Pellet was resuspended with 100 µL FACS Buffer + 100 µL 1% PFA in 1x PBS solution.

**Probe construction, biotinylation and conjugation**
The plasmid constructs for NiVop08 F (pre-F, derived from the Malaysian strain of Nipah virus) were modified by the addition of the sequence encoding the Avi-tag signal for biotinylation (RER-LNDI-FEAQKIEWHE-RER) at the 3′ end of the gene, and the modified genes

were subcloned into the CMV/R expression vector. After expression and purification, the proteins were biotinylated using the BirA biotin-protein ligase reaction kit (Avidity, #BirA500). Biotinylation of the NiV protein probes was confirmed by biolayer interferometry by testing the ability of the biotinylated protein to bind to streptavidin sensors. Retention of antigenicity was confirmed by testing biotinylated proteins against monoclonal antibody 5B3 and/or pool sera from animals immunized with antigen. To conjugate protein probes with the streptavidin-fluorochrome reagents, in a stepwise process, 1/5 of the molar equivalent of the streptavidin-fluorochrome reagent was added to the biotinylated NiVop08 F (pre-F) at 20 min intervals until the molar ratio of streptavidin-fluorochrome reagent: biotinylated protein reached 1:1. The incubation was carried out at 4 °C with gentle rocking. Streptavidin-allophycocyanin (SA-APC) (Invitrogen, #S32362) was mixed with biotinylated NiVop08 F (pre-F).

### Isolation of antigen-specific memory B cells by fluorescence-activated cell sorting (FACS)

The splenocytes we sorted were gated for antigen-specific memory B cells, specifically live IgG+ B cells, which were also positive for the NIV prefusion F-APC probe (live, B220+, CD4-/CD8-/F4/80-/Gr1-, IgM-, IgG+ and NiV prefusion F+). We index sorted 8 96-well plates using a FACS Aria II (BD Biosciences) interfaced with FacsDiva software (BD Biosciences, version 8.0.1). Flow cytometry analyses was performed using FlowJo software (Tree Star, Inc., version 9.9.4).

### Isolation of NiV F-specific B cell sequences

Probe-positive mouse IgG B cells were sorted individually into 96-well PCR plates and stored at −80 °C. All primer sequences used are consistent with those previously published[61] and included in Supplementary Table 4. The first strand cDNA for the expressed H and L chain was synthesized using a commercially available Superscript III reverse transcription kit (Invitrogen, #12574018). The 25 µL reverse transcription reaction mixture consisted of 1x PCR buffer, 1 µM of IgG and IgK antisense primers, 1 mM dNTP mix, 0.5% IGEPAL, 7 fM DTT, 1U of RNase inhibitor, 1U of SuperScript III reverse transcriptase. The reaction was performed as follows; 42 °C for 10 min, 25 °C for 10 min, 50 °C for 60 min, 94 °C for 5 min. This cDNA reaction was used for the 1st round PCR of IgG chains using a commercially available HotstarTaq Plus DNA polymerase kit (Qiagen, #203603). The 10 µL reaction mixture consisted of 1x PCR buffer, 0.2 µM of external mouse IgG primers, 0.2 mM dNTP mix, 1 mM MgCl, 0.5U Taq DNA Polymerase and 1 µL of cDNA reaction. The touchdown PCR reaction was performed as follows; 95 °C for 5 min, [94 °C for 30 sec, 62 °C for 30 sec, 72 °C for 60 sec > 5x], [94 °C for 30 sec, 60 C for 30 sec, 72 °C for 60 sec > 2x], [94 °C for 30 sec, 55 °C for 30 sec, 72 °C for 60 sec > 2x], [94 °C for 30 sec, 52 °C for 30 sec, 72 °C for 60 sec > 25x], [94 °C for 30 sec, 56 °C for 30 sec, 72 °C for 60 sec > 10x], 72 °C for 10 min. This 1st round IgG PCR reaction was used for the 2nd round PCR of IgG chains using a commercially available HotstarTaq Plus DNA polymerase kit. The 10 µL reaction mixture consisted of 1x PCR buffer, 0.2 µM of internal mouse IgG primers, 0.2 mM dNTP mix, 1x Q solution, 0.5U Taq DNA Polymerase and 0.5 µL of 1st round PCR reaction. The touchdown PCR reaction was performed as follows; 95 °C for 5 min, [94 °C for 30 sec, 62 °C sec for 30 sec, 72 °C for 60 sec > 5x], [94 °C for 30 sec, 60 °C for 30 sec, 72 °C for 60 sec > 2x], [94 °C for 30 sec, 55 °C for 30 sec, 72 °C for 60 sec > 2x], [94 °C for 30 sec, 52 °C for 30 sec, 72 °C for 60 sec > 25x], 72 °C for 10 min. The same cDNA reaction was also used for the 1st round PCR of IgK chains using a commercially available HotstarTaq Plus DNA polymerase kit. The 10 µL reaction mixture consisted of 1x PCR buffer, 0.2 µM of external mouse IgK primers, 0.2 mM dNTP mix, 1 mM MgCl, 0.5U Taq DNA Polymerase and 1 µL of cDNA reaction. The PCR reaction was performed as follows; 95 °C for 5 min, [94 °C for 30 sec, 50 °C for 30 sec, 72 °C for 55 sec > 50x], 72 °C for 10 min. This 1st round IgK PCR reaction was used for the 2nd round PCR of IgK chains using a

commercially available HotstarTaq Plus DNA polymerase kit. The 10 µL reaction mixture consisted of 1x PCR buffer, 0.2 µM of (21) internal mouse IgG primers, 0.2 mM dNTP mix, 1x Q solution, 0.5U Taq DNA Polymerase and 0.5 µL of 1st round PCR reaction. 5 µL of both IgG and IgK reactions were analyzed using DNA electrophoresis and the remaining 5 µL was used for Sanger sequencing (ACTG Inc.) with the respective internal primers for 2nd round reactions. Sequences were analyzed via IMGT/HighV-QUEST and in-house post-processing tools. Selected mouse IgG and IgK chains were cloned into humanized IgH and IgL vectors (Genscript).

### Generation of fabs

Fabs were constructed by inserting GLVPRGSHHHHHHH** between PKSCDK and THTCPPCP in the IgG1 hinge region in the heavy chain IgG1 plasmids. Antibody sequences are listed in Supplementary Table 2.

### Antibody expression

Antibodies were expressed by transfection of heavy and light chain plasmids together (or Fabs and light chain plasmids) into Expi293F cells grown in suspension at 37 °C using Turbo293 (SPEED Biosystem, #PX1002). Five days post-transfection, the supernatant was collected and centrifuged, protease inhibitors were added to clarified supernatant and purified using protein A-agarose resin (ThermoFisher, #20334). Bound antibody is eluted with IgG elution buffer into 1/10th volume of 1 M Tris-HCl (pH 8.0).

### Biolayer interferometry

A fortéBio Octet Red384 instrument was used to measure antibody binding to NiV prefusion and postfusion F proteins. The protein sequences (Supplementary Table 1), quality and structural integrity of the prefusion F and postfusion F proteins used in biolayer interferometry binding studies are described in Loomis et al., 2020[42]. The fortéBio Octet Red384 instrument obtains real-time kinetic binding data[62]. His1K biosensors (fortéBio, #18-5122) were equilibrated for >600 sec in 1x PBS prior to loading with his-tagged NiV pre-F or NiV post-F protein (25 µg/mL in Blocking Buffer (1x PBS, 1% BSA (Sigma-Aldrich, #A9576-50mL)) for 600 sec. Following loading, sensors were incubated for 60 sec in Blocking Buffer prior to incubation with the mAb (25 µg/mL in Blocking Buffer) for 600 sec. Sensors were then incubated in Blocking Buffer for 600 sec. Parallel correction to subtract systematic baseline drift was carried out by subtracting the measurements recorded for a loaded sensor incubated in 1x PBS + 1% BSA. Data analysis and curve fitting were carried out using Octet Data Analysis v12.0.2.3. Experimental data were fitted with the binding equations describing a 1:1 interaction. Global analysis of the data sets assuming reversible binding (full dissociation) were carried out using nonlinear least-squares fitting allowing a single set of binding parameters to be obtained simultaneously for all of the concentrations used in each experiment. All the assays were performed with agitation set to 1,000 rpm at 30 °C, in 1x PBS supplemented with 1% BSA to minimize nonspecific interactions with a final volume of 60 µL/well.

### Measurements of antibody (Fab) affinity to prefusion NiV F and HeV F by biolayer interferometry

Purified prefusion NiV F and prefusion HeV F were diluted to 25 nM in BLI affinity buffer (150 mM NaCl, 10 mM HEPES pH 8, 0.05% (v/v) Tween 20 and 1 mM EDTA supplemented with 1 mg/mL BSA). Fabs were diluted to 100 nM working concentration in BLI affinity buffer, then diluted in series in a 96 well plate. For determining binding affinity, prefusion NiV F was adhered to streptavidin-coated biosensor tips (loading step), then dipped in parallel into a dilution series of Fabs (association step) and finally BLI affinity buffer (dissociation step). Raw association and dissociation curves were reference subtracted and fit to a 1:1 binding model. For comparison of antibody binding to

prefusion NiV F and HeV F, F proteins were adhered to streptavidin-coated biosensor tips and then dipped into a panel of Fabs, each at 50 nM concentration ($\geq$10-fold above the $K_D$ determined from previous experiments), followed by a dissociation step into BLI affinity buffer.

## Immunogenic characterization of isolated NiV F monoclonal antibodies

A pseudovirus neutralization assay was used because NiV is classified as a BSL-4 pathogen. Neutralizing antibody titers were determined using a microneutralization assay using VSV$\Delta$G-luciferase expressing NiV F and NiV G/RBP in Vero E6 cells (ATCC, #CRL-1587) as previously described[42,43]. NiV F/G VSV$\Delta$G-luciferase pseudovirus was first incubated with anti-VSV G monoclonal antibody (8G5, Kerafast, #EB0010) for 15 min to neutralize any trace infection due to residual VSV G that may have been incorporated into the particles pseudotyped with NiV F and G/RBP proteins. Each monoclonal antibody was serially diluted in DMEM with 10% FBS, 1% Pen/Strep, 1% GlutaMax and mixed with equal volume of pseudotyped particles with anti-VSV G antibody, incubated for 30 min at room temperature before addition to Vero E6 cells. After 24 hours, medium was removed by aspiration. Cell lysis (Promega, #E1531) and detection of firefly luciferase (Promega, #E1501 or #E4550) were performed according to the protocol recommended by the manufacturer. Briefly, firefly luciferase assay lysis buffer was thawed to room temperature, diluted 1:5 with ddH$_2$O and 20 $\mu$L was added to each well. Plates were placed on an orbital shaker for 20-30 min. Following lysis, 50 $\mu$L of luciferase assay reagent was added to each well and read at 570 nm on the SpectraMax L luminometer (Molecular Devices). Percent neutralization was normalized considering uninfected cells as 100% neutralization and cells infected with only pseudovirus as 0% neutralization. The 80% inhibitory concentration (IC$_{80}$) was calculated by curve fitting and non-linear regression (log(agonist) vs normalized response (variable slope) EC) in triplicate wells using GraphPad Prism v8.

## Competitive NiV mAb binding assay using biolayer interferometry

Antibody cross-competition was determined based on biolayer interferometry using a fortéBio Octet HTC instrument. His1K biosensors (fortéBio, #18-5122) were equilibrated for >600 sec in 1x PBS prior to loading with his-tagged NiV prefusion F (25 $\mu$g/mL in Blocking Buffer (1x PBS, 1% BSA) for 600 sec. Following loading, sensors were incubated for 60 sec in Blocking Buffer prior to incubation with the competitor mAb (25 $\mu$g/mL in Blocking Buffer) for 600 sec. Sensors were then incubated in Blocking Buffer for 600 sec before incubation with analyte mAb (25 $\mu$g/mL in Blocking Buffer) for 600 sec. Percent competition (PC) of analyte mAbs binding to competitor-bound NiV prefusion F was determined using the equation: PC = 100 – [(analyte mAb binding in the presence of competitor mAb) / (analyte mAb binding in the absence of competitor mAb)] x 100. All the assays were performed with agitation set to 1,000 rpm at 30 °C. For all competition experiments, we ensured that antibodies had very slow off rates to mitigate potential artifacts that might have arisen from antibody dissociation. Experiments were performed with IgGs (not Fabs), making use of the intrinsic avidity of bivalent IgGs. A 60% threshold was used to determine competition, consistent with previously published studies[63].

## Cryo-EM sample preparation and data collection

Purified NiV F and Fabs were mixed at a molar ratio of 1:1.2 (NiV-F:Fab) in 2 mM Tris pH 8, 200 mM NaCl, 0.02% (w/v) sodium azide and 0.1% (w/v) amphipol A8-35. The final concentration of NiV F in each sample ranged from 0.5-0.7 mg/mL, or approximately 3-5 $\mu$M. The samples were incubated for 30 minutes at room temperature to form complexes, then centrifuged at 20,000 x $g$. Approximately 3-4 $\mu$L of supernatant was deposited onto plasma-cleaned UltrAuFoil R 1.2/1.3,

300 mesh grids (Electron Microscopy Sciences). Excess liquid was blotted away for 3-4 seconds, and the grids were plunge frozen in liquid ethane using a Vitrobot Mark IV (FEI) operating at 22 °C and 100% humidity. Frozen grids were stored in liquid nitrogen until further use.

Grids were screened for quality using a Talos F200C (FEI) transmission electron microscope (TEM). Grids that passed quality control were loaded onto a separate microscope for high resolution imaging. All high-resolution structures were imaged on a Titan Krios TEM (ThermoFisher Scientific) operating at 300 kV equipped with a K3 camera (Gatan), except for prefusion NiV F complexed with Fab 1H1, which was imaged on a Glacios TEM equipped with a Falcon 4 detector (ThermoFisher Scientific). Movies were collected using SerialEM. Calibrated pixel sizes were 1.075 Å (1H8 and 4H3), 0.66 Å (2D3 and 2B12), 0.94 Å (prefusion NiV+1H1), and 0.81 Å (1A9).

## Cryo-EM data processing and model building

Motion correction and CTF-estimation were performed in WARP or cryoSPARC Live. Micrographs were imported into cryoSPARC[64] for particle picking, 2D classification, ab initio 3D reconstruction and 3D refinement[65]. Diagrams of the cryo-EM data processing and workflows are shown in Supplementary Figures 3-8. Local resolution estimates are shown in Supplementary Figure 9. The initial model for NiV F was PDB ID 6TYS. Homology models for the Fabs were generated using ABodyBuilder[66]. Initial models were docked into the cryo-EM maps using Chimera. The complementarity-determining loops were built manually in Coot. Models were iteratively refined using Coot, Phenix and ISOLDE[67,68]. Cryo-EM data collection and refinement statistics are shown in Supplementary Table 3.

## Surface plasmon resonance (SPR)

Purified NiV F was immobilized to a single flow cell of either (i) a CM5 sensor functionalized with an anti-StreptagII antibody or a NiNTA sensor in a Biacore X100 (GE Healthcare). Binding kinetics for Fabs 4H3 and 1H8 were measured using a CM5 sensor, whereas kinetics for Fabs 2D3, 1A9, 1H1 and 2B12 were measured using a NiNTA sensor. The running buffer was 1x HBS-P + at pH 8. Serial two-fold dilutions of Fabs 4H3, 2D3, 1H8, 1A9, 1H1 and 2B12 were injected over both the reference cell and the functionalized flow cells. Data were double reference-subtracted and fit to a 1:1 binding model using Biacore Evaluation software.

## Reporting summary

Further information on research design is available in the Nature Portfolio Reporting Summary linked to this article.

## Data availability

Structural models are deposited in the protein data bank (PDB, https://www.rcsb.org/).The PDB IDs are: 7UOP, 7UP9, 7UPA, 7UPK, 7UPB, 7UPD. Cryo-EM maps are deposited in the EM Database (https://www.emdataresource.org/). The maps are publicly available and can be found using the following EMD IDs: 26652, 26658, 26659, 26668, 26660, 26662. All data supporting the findings of this study are within the article and its Supplementary Information files or provided as a Source Data file. Source data are provided with this paper.

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

## Acknowledgements

We thank Sasha Dickinson, Nicole Johnson, and Jory Goldsmith for technical assistance with grid loading and microscope operation. We thank John Ludes-Meyers and Kaci Erwin for assistance with mammalian protein expression. We thank Maria Cantelli for administrative support and members of the NIH NIAID VRC Translational Research Program for technical assistance with mouse experiments. We thank Madeline Sponholtz and Scott Rush for their careful readings of the manuscript. This work was supported in part with federal funds from the Frederick National Laboratory for Cancer Research, NIH, under contract HHSN261200800001 (Y.T.), and also by The Welch Foundation grant number F-0003-19620604 (J.S.M).

## Author contributions

R.J.L., P.O.B., J.S.M. and B.S.G conceived the project. P.O.B, B.E.F, D.R.A., E.G.B., Y.T. and R.J.L performed studies and analyzed the results. P.O.B. and R.J.L. wrote the original draft and P.O.B., B.S.G., J.S.M. and R.J.L. revised and edited the manuscript. J.S.M., B.S.G. and R.J.L. supervised the studies.

## Competing interests

Y.T. is employed by Leidos Biomedical Research, Inc., supported in part with funds from the Frederick National Laboratory for Cancer Research, NIH, under contract HHSN261200800001. B.S.G. and R.J.L. are inventors on US patent application #63315934, filed March 2, 2022, entitled "Monoclonal Antibodies to Nipah Virus F Protein and Their Use". This patent describes the 10 neutralizing antibodies in this manuscript. The remaining authors declare no competing interests.
