## [Peer Review File · Nature Communications]

Reviewers' Comments:

Reviewer #1:

Remarks to the Author:

Byrne et al presents a comprehensive structural epitope map of antibodies targeted against the pre-fusion conformation of NiV-F. The structure of pre-fusion NiV-F as well as the structural basis for several nAbs targeted against pre-fusion NiV-F has been previously elucidated. However, there is value in the comprehensive characterization of anti-NiV-F structural epitopes presented in this paper.

Too often, structural epitope papers focus on the authors' own antibodies without direct comparisons with other published antibodies. Here, the authors contextualize their Mab epitopes and potency (at least against NiVpp) with previously published anti-NiV-F MAbs (mAb066 and h5B3). I consider this a strength, which can be further improved (see comments below). In addition, they identify novel neutralizing epitopes in domains I and II that involve interesting quaternary epitopes (1H1, 1A9, 2B12, 4F6). Using pre-fusion stabilized NiV-F as the immunogen and biotinylated stabilized pre-fusion NiV-F as bait for sorting antigen-specific B cells, the authors identified several dozen MAbs that mapped into 8 neutralizing epitopes, 4 of which have not been described. Surprisingly, novel neutralizing epitopes span the basal and lateral sides of trimer. Previous studies suggest that the trimer apex formed by domain III was immunodominant.

While one can always request live virus studies and ever more detailed functional and mechanistic studies, I believe the sheer number of structural epitopes identified would be of interest to the greater community. Six high resolution (<3 angstroms) and four moderate resolution (4-6 angstroms) structures of Fab-F trimer complexes that yield biologically novel and interesting features compensate somewhat for the relative paucity of functional data.

Thus, in the absence of more functional data, the authors should do their utmost to broaden the impact of their structural biology data. My comments are restricted to how their structural data can be presented with greater clarity and in a form that would be of practical use to the greater community.

1. Why is there such a long wait between the second and third boost at week 10 and 34, respectively (Fig. 1A)? Was this a purposeful strategy or was the project interrupted by the pandemic? This should be stated either way so that the timeline for the "vaccination" strategy can be properly understood. One should be transparent about the rationale whatever the cause.
2. IC80 values were calculated for the 10 mAbs under study (Methods, lines 486-488) but nowhere are the values indicated in the text. The authors should list the IC80 and IC50 values of all the MAbs tested either alongside Fig. 1D or as a separate Table in the main text (especially in comparison with Mab066 and h5B3, previously published Abs used as positive controls).
3. Fig. 1D & Fig. 2 (and Supplementary Fig. 2): tightest binders (lowest Kds) are not always the most potent neutralizers. For example, 1H8 and 1A9 have much lower Kds (<20 pM and 1.7 nM, respectively; Fig. 2C and 2D) than 4H3 (5.9 nM; Fig. 2A), but appear to have IC50s up to 10-fold higher (once again, a Table of IC50/IC80 would be very useful). Similarly, F2 and 4F6 have Kd <1 nM but have IC50 >10-fold more higher 4H3. A table listing their putative IC50/IC80 values and their measured Kds would allow the readers to better appreciate the gestalt of the relationship (or lack thereof) between binding affinity, epitope, and neutralization potency. Any such relationship (or lack thereof) should be commented on in the discussion. A single paper does not have to solve every mystery; the strength of this paper lies in its comprehensive delineation of neutralizing epitopes. Transparent presentation of the data can broaden the impact of the study.
4. Fig. 3 and 4 (line 134-138): 4H3 is said to bind to an epitope that involves one hydrogen bond to a glycan at Asn 67. It is not clear where the Asn67 glycan interaction is shown in either Fig. 3A or 4A. Can the heterogenous glycan be visualized in their cryoEM maps? What glycan moiety if present? This should be commented on.
5. Fig. 5: This is a potentially important figure for the field. The authors should make this figure as clear and useful as possible. The accession numbers of the henipavirus sequences used to generate this conservation map should be included in the figure legend or in a separate methods section. These should include at least the ref strains of NiV-Malaysia, NiV-Bangladesh, HeV, HeV-g2 variant (Genbank: MZ318101), Ghana virus (GhV aka GhM74a) and Cedar virus (CedV). ICTV

will soon reclassify Mojiang virus (MojV) as not being a henipavirus, and so MojV and all its recent relatives may be ignored. I strongly suggest that the multiple sequence alignment used to generate this figure be shown as a supplementary figure.

5a. There are 19 divergent HNV clades in Africa, only one of which also has its fusion and attachment glycoprotein sequences available and their protein functionally characterized (the aforementioned Ghana virus). GhV and CedV is as divergent from each other as they are from NiV/HeV.

5b. Understanding the degree of conservation in the F proteins between these extant HNV species is important to the stated goal of the current study.

6. Related to Fig.5. (lines 174-176) The authors states "a multiple sequence alignment of F proteins within the Henipavirus genus indicates that neutralizing antibodies reported here recognize pockets of conserved surface residues." Would MAbs that recognize epitopes in these conserved regions be predicted to bind or neutralize other HNVs (CedV, GhV, and HeV)? For example, 2B12 (& 4F6) and 1H1 appears to bind particularly conserved epitopes relative to the other Abs (Fig. 5D) – would these antibodies be predicted to bind or neutralize more divergent HNVs? The authors should perform the analysis and comment on whether these conserved regions at the basal surface of the pre-fusion HNV-F serve as a target for the most broadly neutralizing Abs.

6a. Once again, in the absence of functional evidence, a rational interpolation of their functional data would do a great service for the field of paramyxovirus fusion and vaccine development.

MINOR COMMENTS:

1. CTV has recommended that RBP (receptor binding protein) be used to replace HN/H/G for paramyxoviruses (Rima et al, 2019; PMID: 31609197). The authors should define the old to new nomenclature once in the abstract and introduction, and then refer to NIV-G as NiV-RBP thereafter.

2. The authors should specify that NiVop08 is derived from the NIV- Malaysia strain at least once.

3. What do the asterisks mean in Fig. 1C and Fig. S1F?

4. One sees that 2C2 is a non-neutralizing antibody (Suppl Fig. 1F), but this should be noted in the legend to Fig. 1D for clarity.

5. Line 36-38: Ref 4 reported only on 94 patients (3 of which had non-encephalitic NIV infections). Not all NIV infections result in encephalitis although all NiV-related deaths are from encephalitis. The figures cited in the authors' text (265 *infections* resulting in 105 deaths and one million pigs culled) were from the ultimate summaries of the NiV outbreak in Malaysia (e.g. Chua KB et al, Science 2000;288(5470):1432-1435) and governmental reports. Please use either the Chua et al ref or the henipavirus chapter in latest edition of Fields Virology (7th edition, 2020).

6. 53 Mab were derived from paired Heavy and Light chains. 21 had strong preference for pre-fusion NiV-F, 14 bound both pre and post-fusion NiV-F. What happened to the remaining 18? Please comment for clarity.

Reviewer #2:

Remarks to the Author:

The manuscript by Byrne and colleagues described 10 potent neutralizing mAbs against Nipah virus fusion protein (NiVF) biochemically and structurally, including 4 newly identified epitopes. In this study, the authors immunized mice with prefusion-stabilized NiVF and sorted 10 neutralizing antibodies with 9 of which bound to prefusion NiVF only. All antibodies' affinity have been biochemically described, and the corresponding epitopes have been biochemically and structurally described using competition BLI as well as cryo-EM through NiVF-Fab complex. Among all the structures, 6 were obtained at high resolution and 4 were at moderate resolution, which described epitopes across the F protein including lateral and basal epitopes.

Although among 10 mAbs 6 of which the corresponding epitopes overlap with previously described mAbs, to observe these epitopes once again in independent study really enhanced the importance of these potent neutralizing epitopes. On the other hand, the description of 4 novel epitopes, especially the 2 basal epitopes, really allows us to get a more complete picture of NiVF antigenicity.

While the study is interesting and the result is indeed important, there are still reasonable amount of manuscript and experimental details need to be further addressed:

Major:

1. Please provide the sequences for all mAb mentioned in the paper as a supplementary figure/table. Data disclosure is necessary for pushing the field forward.
2. For all biochemical experiments, especially the BLI/SPR data, please indicate the number of replicates. If no replicate has been done, please conduct biological replicates if possible or at least technical replicates. Also please show the statistical analysis when applicable.
3. Please provide the postfusion NiVF construct used in this study together with QC for both protein quality and conformation.
4. For all competition BLIs, please address whether a fast Kon/Koff antibody will be fully covered. Since based on the method section, the dissociation step may wash out such antibodies.
5. Figure 1D: Since the SD for each data point is relatively large, please provide the statistical analysis (R2 for example) for the curve fitting.
6. Figure 1E: Can the authors explain why use "60%" as a background threshold, which is pretty high. Also please address where these background competitions may come from.
7. Line 157: Since "apical", "lateral" and "basal" have been mentioned multiple times, it would be helpful to define the boundaries of these three regions either on F directly or mAb groups (for example, based on Figure 1E).
8. Line 175/Figure 5B-D: Please provide the MSA result. It seems the lateral surface is not that conserved (though most variations facing inside, there still quite a lot of non-conserved residues on the surface), can the authors make some comments on this?
9. Figure S1E: Please explain what "RU" refers to here as well as how and why to choose this value for BLI data.

Minor:

1. Line 60-61: The authors mentioned fusion inhibition antibodies for both F and G proteins, but only cited mAb work for F protein.
2. Figure 4A/Line 137: In line 137 Asn67 has been mentioned, but in panel A Asn77 has been shown.
3. Line 148: Since for all the other mAbs the number of H-bond has been described, it would be appreciated if the number can be addressed for 1A9 as well.
4. Line 223-224: Please cite the corresponding figure properly.
5. Figure 1C: Please indicate what "*" means in the legend.
6. Figure 1E: The thickness of the lines inside the matrix is not intuitive. Please explain the meaning of the thicker lines in the legend.
7. Figure 3A-C,F: Since the volume data has been shown, it would be better to hide the ribbon representation for F and Fab, as D&E. And mentioning how many Fab densities have been observed for each map in the legend will be easier for the reader.

To the reviewers: Thank you for your responses and thoughtful consideration of our manuscript.

REVIEWER COMMENTS

Reviewer #1 (Remarks to the Author):

Byrne et al presents a comprehensive structural epitope map of antibodies targeted against the pre-fusion conformation of NiV-F. The structure of pre-fusion NiV-F as well as the structural basis for several nAbs targeted against pre-fusion NiV-F has been previously elucidated. However, there is value in the comprehensive characterization of anti-NiV-F structural epitopes presented in this paper.

Too often, structural epitope papers focus on the authors' own antibodies without direct comparisons with other published antibodies. Here, the authors contextualize their Mab epitopes and potency (at least against NiVpp) with previously published anti-NiV-F MABs (mAb066 and h5B3). I consider this a strength, which can be further improved (see comments below). In addition, they identify novel neutralizing epitopes in domains I and II that involve interesting quaternary epitopes (1H1, 1A9, 2B12, 4F6). Using pre-fusion stabilized NiV-F as the immunogen and biotinylated stabilized pre-fusion NiV-F as bait for sorting antigen-specific B cells, the authors identified several dozen MABs that mapped into 8 neutralizing epitopes, 4 of which have not been described. Surprisingly, novel neutralizing epitopes span the basal and lateral sides of trimer. Previous studies suggest that the trimer apex formed by domain III was immunodominant.

While one can always request live virus studies and ever more detailed functional and mechanistic studies, I believe the sheer number of structural epitopes identified would be of interest to the greater community. Six high resolution (<3 angstroms) and four moderate resolution (4-6 angstroms) structures of Fab-F trimer complexes that yield biologically novel and interesting features compensate somewhat for the relative paucity of functional data.

Thus, in the absence of more functional data, the authors should do their utmost to broaden the impact of their structural biology data. My comments are restricted to how their structural data can be presented with greater clarity and in a form that would be of practical use to the greater community.

Response: Thank you for the accurate summary of the paper and suggestion to broaden the impact of our structural biology data. We have revised the manuscript to focus on that impact. Changes to the manuscript are indicated with their line numbers in this document, set apart from the rest of our response with half-inch indentations. When possible, we reproduced the changes to the manuscript in their entirety.

We note that the Line numbers listed below correspond to our word document with the track changes toggled "OFF". We preserved the tracked changes, which can be toggled on for details about the changes but which can cause errors in the display of line numbers.

1. Why is there such a long wait between the second and third boost at week 10 and 34, respectively (Fig. 1A)? Was this a purposeful strategy or was the project interrupted by the

pandemic? This should be stated either way so that the timeline for the “vaccination” strategy can be properly understood. One should be transparent about the rationale whatever the cause.

Response: Thank you for this comment. We have revised the Results (Lines 88-91) and Methods (Lines 463-469) to reflect the response below. The animals used for the B-cell sorting were immunized in a prime-boost-boost strategy (wk 0, 3, 10) to evaluate immunogenicity of our NiV prefusion F antigen designs. We waited about 6 months between initial immunization series (wk 0-wk10) and the boost used to sort B cells (wk 34, 42) to allow for maturation and increased affinity of the antibody response. Following assay optimization and serum analysis (reported in Loomis et al, *Frontiers in Immunology* 2020), we boosted the animals (week 34) to perform probe titration and a B-cell sort in the desired 10-20 day peak response following boost. Our first sort did not yield the percentage of IgG positive single cells we were expecting due to a generous IgG vs IgM gating strategy. We waited for antibody response to start to wane, boosted again at week 42 and used a less generous IgG vs IgM gating strategy.

The changes are reproduced here:

Results (Lines 88-91): “The mice were boosted at week 34 and 42 following an initial immunization series (week 0, 3 and 10). The time between the initial immunization series and the boosts used for B cell sorting allowed for maturation and increased affinity of the antibody response.”

Methods (Lines 463-469): “We boosted mice at week 34 to perform probe titration and B cell sorting in the desired 10-20 day peak response following boost. Our first sort did not yield the percentage of IgG positive single cells we were expecting due to a generous IgG vs IgM gating strategy. We waited for the antibody response to start to wane, boosted again at week 42 and used a less generous IgG vs IgM gating strategy. The interval between the B cell sorting boosts (week 34 and 42) and the initial immunization series (week 0, 3 and 10) allow for maturation and increased affinity of the antibody response.”

2. IC80 values were calculated for the 10 mAbs under study (Methods, lines 486-488) but nowhere are the values indicated in the text. The authors should list the IC80 and IC50 values of all the MAbs tested either alongside Fig. 1D or as a separate Table in the main text (especially in comparison with Mab066 and h5B3, previously published Abs used as positive controls).

Response: Thank you for bringing this omission to our attention. We have added a table that lists the IC80 values for all of the mAbs tested with values determined for 066 and h5B3 in Table 1 (main text). This Table also includes the Kd values for Fab binding to F protein determined by SPR and BLI, to address an additional comment (#3) by this reviewer. IC80 values were consistent with previously published results (Dang et al 2019, Avanzato et al 2019).

3. Fig. 1D & Fig. 2 (and Supplementary Fig. 2): tightest binders (lowest Kds) are not always the most potent neutralizers. For example, 1H8 and 1A9 have much lower Kds (<20 pM and 1.7

nM, respectively; Fig. 2C and 2D) than 4H3 (5.9 nM; Fig. 2A), but appear to have IC50s up to 10-fold higher (once again, a Table of IC50/IC80 would be very useful). Similarly, F2 and 4F6 have $K_d < 1$ nM but have IC50 >10-fold more higher 4H3. A table listing their putative IC50/IC80 values and their measured K_d s would allow the readers to better appreciate the gestalt of the relationship (or lack thereof) between binding affinity, epitope, and neutralization potency. Any such relationship (or lack thereof) should be commented on in the discussion. A single paper does not have to solve every mystery; the strength of this paper lies in its comprehensive delineation of neutralizing epitopes. Transparent presentation of the data can broaden the impact of the study.

Response: This is a great point regarding what characteristics are used to select antibodies for further evaluation. We have not found a correlative relationship between binding affinity, epitope location, and neutralization potency, however we have compiled the K_d values for Fab binding to NiV F protein along with the IC80 values in Table 1 (main text) and added a plot of IC80 vs K_D in Supplementary Figure 2.

We have also added the following text to the Results (Lines 136-137) and Discussion (Line 236-238):

“We observed no correlation between binding affinity and neutralization potency”.

“Interestingly, the binding affinity of NiV F Fabs does not correlate with their neutralization potency, which suggests a complex relationship between antibody binding and F protein function during fusion.”

We also added additional plots of IC80 vs binding affinity (Supplementary Figure 2E).

4. Fig. 3 and 4 (line 134-138): 4H3 is said to bind to an epitope that involves one hydrogen bond to a glycan at Asn 67. It is not clear where the Asn67 glycan interaction is shown in either Fig. 3A or 4A. Can the heterogeneous glycan be visualized in their cryoEM maps? What glycan moiety if present? This should be commented on.

Response: The map quality at Asn67 is sufficient to model the first glycan moiety (GlcNAc) in the chain, however beyond that we cannot build it with much confidence. It is possible that this first glycan forms an interaction, either Van der Waals or hydrogen bond (or both), however in light of the reviewer’s comment, we have removed our statement that this glycan moiety is involved in a hydrogen bond interaction. If this glycan were ordered by antibody binding, then it seems likely that additional sugars might also be ordered. Since they are not, we opt to simply note its proximity to the epitope and note that the map quality did not permit modeling of an interaction. We have made the following addition to the text (Lines 150-152):

“The epitope abuts the glycan at Asn67, however the map quality was insufficient in this region to model an interaction between 4H3 and the glycan itself.”

5. Fig. 5: This is a potentially important figure for the field. The authors should make this figure as clear and useful as possible. The accession numbers of the henipavirus sequences used to generate this conservation map should be included in the figure legend or in a separate

methods section. These should include at least the ref strains of NiV-Malaysia, NiV-Bangladesh, HeV, HeV-g2 variant (Genbank: MZ318101), Ghana virus (GhV aka GhM74a) and Cedar virus (CedV). ICTV will soon reclassify Mojiang virus (MojV) as not being a henipavirus, and so MojV and all its recent relatives may be ignored. I strongly suggest that the multiple sequence alignment used to generate this figure be shown as a supplementary figure.

5a. There are 19 divergent HNV clades in Africa, only one of which also has its fusion and attachment glycoprotein sequences available and their protein functionally characterized (the aforementioned Ghana virus). GhV and CedV is as divergent from each other as they are from NiV/HeV.

5b. Understanding the degree of conservation in the F proteins between these extant HNV species is important to the stated goal of the current study.

Response: We agree that Figure 5 is a potentially important figure for the field and our understanding of how conserved antigenic sites elicit a broadly neutralizing antibody response within a viral family. We have added a list of all the Henipavirus F protein sequences used for the alignments in Supplementary Figure 11, including their GenBank accession numbers in the figure legend.

6. Related to Fig.5. (lines 174-176) The authors states “a multiple sequence alignment of F proteins within the Henipavirus genus indicates that neutralizing antibodies reported here recognize pockets of conserved surface residues.” Would mAbs that recognize epitopes in these conserved regions be predicted to bind or neutralize other HNVs (CedV, GhV, and HeV)? For example, 2B12 (& 4F6) and 1H1 appears to bind particularly conserved epitopes relative to the other Abs (Fig. 5D) – would these antibodies be predicted to bind or neutralize more divergent HNVs? The authors should perform the analysis and comment on whether these conserved regions at the basal surface of the pre-fusion HNV-F serve as a target for the most broadly neutralizing Abs.

6a. Once again, in the absence of functional evidence, a rational interpolation of their functional data would do a great service for the field of paramyxovirus fusion and vaccine development.

Response: Thank you for this comment. We recognize that while there is sequence divergence among Henipaviruses, there are still regions or pockets on the surface of the prefusion protein that are highly conserved across all Henipavirus family members identified and even within the larger paramyxovirus family. It has been shown previously that antibodies to Nipah F and G are cross-reactive to Hendra and can protect against challenge in several animal studies, and the reverse is true. From the data we have, we would speculate that these identified antibodies would not only bind Hendra F but also neutralize Hendra infection. We have included data showing binding of these antibodies to Hendra prefusion F protein in Supplementary Figure 12.

We attempted to express Cedar virus F protein by transferring the prefusion-stabilizing mutations from NiV F to CedV F, however the protein expressed poorly and did not appear fully stabilized in the prefusion conformation (Loomis et al 2020 Frontiers in Immunology). This may be due to the low overall sequence conservation between NiV and CedV. Indeed, the sites surrounding the disulfide and cavity-filling substitutions in NiV F (L172F, L104C/I114C) align especially poorly with CedV F and GhV F

(Supplementary Figure 11). In the absence of functional data, we generated homology models of the MojV, CedV, and GhV F proteins and aligned them with the prefusion NiV F protein, and they do not appear to clash with the antibodies. We are not showing this information in the text however, since we do not place too much weight on this evidence from a template-based homology model. Further work will be required to engineer prefusion-stabilized forms of the MojV, CedV, and GhV F proteins.

The modified text begins on Line 190:

“...suggesting that neutralizing antibodies elicited by NiV F might also recognize related *Henipavirus* F proteins.

To investigate the breadth of antibody recognition across viruses, we compared binding of neutralizing antibodies to both NiV F and Hendra virus (HeV) F. We chose a subset of antibodies from our study that bind the apical, lateral, or basal face of NiV F and assessed binding to HeV F by BLI (Supplementary Figure 12). All antibodies tested bound both NiV F and HeV F. Antibodies directed to the apical and basal faces exhibited no differences in binding to NiV F and HeV F (2D3, 1H8, 1H1, 2B12, 4F6), however two antibodies directed to the lateral face of NiV F (1A9 and 1F2) displayed reduced binding to HeV F when compared to NiV F. Indeed, sequence conservation is greatest in the apical and basal surfaces (Figure 5B-D), suggesting that this region may better tolerate antigenic drift. On the contrary, antibody 1H8 binds the lateral face but shows equal preference for NiV F and HeV F. Antibodies that recognize the basal face (1H1 and 2B12) exhibit especially high sequence conservation (Supplementary Figure 11).

We attempted to produce a prefusion-stabilized F protein from Cedar virus, which has lower sequence conservation with NiV F, but we were unsuccessful. We transferred the prefusion-stabilizing substitutions from NiV F to CedV F, however the protein expressed poorly and showed low stability [42]. This may be due to the low overall sequence conservation between NiV F and CedV F. Indeed, the residues surrounding the sites of disulfide and cavity-filling substitutions in NiV F (L172F, L104C/I114C) align especially poorly with CedV F and Ghanaian bat virus (GhV) F (Supplementary Figure 11). Further work will be required to engineer prefusion-stabilized forms of the Mòjiāng virus, CedV, and GhV F proteins, construct pseudotyped virus neutralization assays, and experimentally test the breadth of antibody binding and neutralization.”

MINOR COMMENTS:

1. CTV has recommended that RBP (receptor binding protein) be used to replace HN/H/G for paramyxoviruses (Rima et al, 2019; PMID: 31609197). The authors should define the old to new nomenclature once in the abstract and introduction, and then refer to NiV-G as NiV-RBP thereafter.

Response: We were unaware of this change and are glad to learn of it. We have added a note in the text identifying NiV-G as NiV-RBP (Line 21), and we used “NiV G/RBP” thereafter. We think a hybrid denotation (G/RBP) strikes a good balance between remaining consistent with nomenclature used in our previous manuscripts and others

published recently in the field, while also promoting the new nomenclature.

2. The authors should specify that NiVop08 is derived from the NIV- Malaysia strain at least once.

Response: We have revised both the results and methods accordingly.

Results (Lines 86-91): “CB6F1/J mice were immunized with NiV F protein stabilized in its prefusion conformation (previously referred to as NiVop08, derived from Malaysian strain)”

Methods (Lines 494-495): “Probe Construction, Biotinylation and Conjugation. The plasmid constructs for NiVop08 F (pre-F, derived from the Malaysian strain of Nipah virus) were modified ...”

3. What do the asterisks mean in Fig. 1C and Fig. S1F?

Response: The asterisks indicate antibodies that are neutralizing. We have modified the figure legends (Fig 1C and Supplementary Figure 1E) to include this information.

Fig 1C legend (Lines 314-316): “(C) Representative 2D class averages from negative stain EM of prefusion NiV F, alone and complexed with Fabs. Asterisks indicate antibodies that were shown to be neutralizing in the pseudovirus neutralization assay (Methods Section).”

Supplementary Fig 1E legend (Lines 365-367): “(E) Biolayer interferometry specificity binding for expressed monoclonal antibodies. Asterisks indicate antibodies that were shown to be neutralizing via pseudovirus neutralization assay.”

4. One sees that 2C2 is a non-neutralizing antibody (Suppl Fig. 1F), but this should be noted in the legend to Fig. 1D for clarity.

Response: The figure legend for Fig 1D has been modified to clarify that 2C2 is a non-neutralizing antibody.

Fig 1D legend (Lines 316-318): “(D) Neutralization of pseudotyped virus. Error bars show the standard error of the mean for three independent biologic replicates. 2C2 is a non-neutralizing antibody.”

5. Line 36-38: Ref 4 reported only on 94 patients (3 of which had non-encephalitic NIV infections). Not all NIV infections result in encephalitis although all NiV-related deaths are from encephalitis. The figures cited in the authors’ text (265 *infections* resulting in 105 deaths and one million pigs culled) were from the ultimate summaries of the NiV outbreak in Malaysia (e.g. Chua KB et al, Science 2000;288(5470):1432–1435) and governmental reports. Please use either the Chua et al ref or the henipavirus chapter in latest edition of Fields Virology (7th edition, 2020).

Response: We thank the reviewer for reference suggestion. The reference has been modified to Chua et al., Science, 2000. (Line 41).

6. 53 Mab were derived from paired Heavy and Light chains. 21 had strong preference for pre-fusion NiV-F, 14 bound both pre and post-fusion NiV-F. What happened to the remaining 18? Please comment for clarity.

Response: The other 18 antibodies did not bind NiV prefusion F or postfusion F. We have modified the text in the results section to indicate this:

Results (Lines 102-104): “Of the antibodies tested, 21 showed a strong binding preference for prefusion NiV F, whereas 14 antibodies bound to both prefusion and postfusion NiV F (Supplementary Figure 1E) and 18 did not bind to either prefusion or postfusion NiV F.”

Reviewer #2 (Remarks to the Author):

The manuscript by Byrne and colleagues described 10 potent neutralizing mAbs against Nipah virus fusion protein (NiVF) biochemically and structurally, including 4 newly identified epitopes. In this study, the authors immunized mice with prefusion-stabilized NiVF and sorted 10 neutralizing antibodies with 9 of which bound to prefusion NiVF only. All antibodies' affinity have been biochemically described, and the corresponding epitopes have been biochemically and structurally described using competition BLI as well as cryo-EM through NiVF-Fab complex. Among all the structures, 6 were obtained at high resolution and 4 were at moderate resolution, which described epitopes across the F protein including lateral and basal epitopes.

Although among 10 mAbs 6 of which the corresponding epitopes overlap with previously described mAbs, to observe these epitopes once again in independent study really enhanced the importance of these potent neutralizing epitopes. On the other hand, the description of 4 novel epitopes, especially the 2 basal epitopes, really allows us to get a more complete picture of NiV F antigenicity.

While the study is interesting and the result is indeed important, there are still reasonable amount of manuscript and experimental details need to be further addressed:

Major:

1. Please provide the sequences for all mAb mentioned in the paper as a supplementary figure/table. Data disclosure is necessary for pushing the field forward.

Response: Thank you for this comment, we agree. Antibody sequences are now included in Supplementary Table 2, as well as the protein sequences for NiV F in Supplementary Table 1. All sequences are also included in the deposited PDB structures. PDB IDs and EMD IDs are now included in Supplementary Table 3. The text of the Methods Section (“Generation of Fabs”) has been modified to read:

Lines (626-628): “Antibody sequences are listed in Supplementary Table 2.”

2. For all biochemical experiments, especially the BLI/SPR data, please indicate the number of replicates. If no replicate has been done, please conduct biological replicates if possible or at least technical replicates. Also please show the statistical analysis when applicable.

Response: Biolayer interferometry and surface plasmon resonance experiments were each completed using proteins (fusion protein and Fabs) from two biological replicates, in this case independent transient transfections and purifications. Two technical replicates were performed for each biological replicate. We added standard deviations in Table 1 (main text). Statistical analyses for fits of the association and dissociation rate constants are shown above representative SPR and BLI traces in Figure 2 (main text) and Supplementary Figure 2. We also adjusted the Kd figures in the main text to reflect the new mean values in Table 1. For all other biochemical experiments, including BLI (for binding characterization and competition) and pseudovirus neutralization assays, information regarding replicates was added in figure legends.

3. Please provide the postfusion NiV F construct used in this study together with QC for both protein quality and conformation.

Response: Thank you for bringing this oversight to our attention. The postfusion NiV F construct used in this study along with QC (both protein quality and conformation) are included in the previously published manuscript Loomis et al Frontiers in Immunology 2020. We revised the methods section to add this information, including the sequences in Supplementary Table 1:

(Lines 564-566): The protein sequences (Supplementary Table 1), quality and structural integrity of the prefusion F and postfusion F proteins used in biolayer interferometry binding studies are described in Loomis *et al*, 2020 [42].

4. For all competition BLIs, please address whether a fast Kon/Koff antibody will be fully covered. Since based on the method section, the dissociation step may wash out such antibodies.

Response: The antibodies used in the competition binding experiment had very slow off rates. We used IgGs (not Fabs) to further minimize dissociation, making use of the intrinsic avidity of bivalent IgGs. Below is an example of what the raw data looked like for a single plate. There is no appreciable dissociation of the competitor antibody. We added text to this effect:

(Lines 627-630): For all competition experiments, we ensured that antibodies had very slow off rates to mitigate potential artifacts that might have arisen from antibody dissociation. Experiments were performed with IgGs (not Fabs), making use of the intrinsic avidity of bivalent IgGs. A 60% threshold was used to determine competition, consistent with previously published studies [63].

5. Figure 1D: Since the SD for each data point is relatively large, please provide the statistical analysis (R2 for example) for the curve fitting.

Response: The R² curve fitting values have been added to Table 1 (main text).

6. Figure 1E: Can the authors explain why use “60%” as a background threshold, which is pretty high. Also please address where these background competitions may come from.

Response: Competition can be due to both direct causes and indirect causes. We used the 60% value as the threshold to determine competition. If the value was less than 60%, there was no significant competition between the competitor and analyte mAbs. This threshold is consistent with other published manuscripts – Wang et al 2021 Science “Ultrapotent antibodies against diverse and highly transmissible SARS-CoV-2 variants”. Weak competition can be the result of minor binding interference due to use of IgGs in this analysis or indirect competition.

We added a brief explanation in the methods:

(Lines 629-630): “A 60% threshold is used to determine competition, consistent with previously published studies (Wang et al 2021).”

7. Line 157: Since “apical”, “lateral” and “basal” have been mentioned multiple times, it would be helpful to define the boundaries of these three regions either on F directly or mAb groups (for example, based on Figure 1E).

Response: We have included a primary sequence and structure diagram in Figure 1F and noted the relative positions of the apical, lateral and basal faces, which correspond to the pre-existing domain denotations of D3 (apical), D1 (lateral), and D2 (basal). The sequence and domain architecture for Henipaviruses are not intuitive since the regions of disparate primary sequence make up parts of the same domains. We came up with the apical/lateral/basal nomenclature to orient descriptions of the domains relative to the membrane.

8. Line 175/Figure 5B-D: Please provide the MSA result. It seems the lateral surface is not that

conserved (though most variations facing inside, there still quite a lot of non-conserved residues on the surface), can the authors make some comments on this?

Response: We thank the reviewer for their observation. We added domain indicators to the MSA in Supplementary Figure 11, and have noted this in a new sentence at the end of our results section:

(Lines 199-203): “Indeed, sequence conservation is greatest in the apical and basal surfaces (Figure 5B-D), suggesting that this region may better tolerate antigenic drift. On the contrary, antibody 1H8 binds the lateral face but shows equal preference for NiV F and HeV F. Antibodies that recognize the basal face (1H1 and 2B12) exhibit especially high sequence conservation (Supplementary Figure 11).

9. Figure S1E: Please explain what “RU” refers to here as well as how and why to choose this value for BLI data.

Response: We modified RU to “Response (nm)” in units of nanometers in Supplementary Figure 1E, to more closely align with the standard for biolayer interferometry in the literature. The response in nanometers represents the shift in the measured interference pattern in each biosensor tip. We also fixed a related error in Supplementary Figure 12 (changed y-axis from RU to Response). We added more detail in the methods section and referenced a manuscript (Reference 63, Line 630) that more thoroughly describes biolayer interferometry and its analysis.

Minor:

1. Line 60-61: The authors mentioned fusion inhibition antibodies for both F and G proteins, but only cited mAb work for F protein.

Response: Thank you for bringing this oversight to our attention. We have modified the cited work to include a recent paper by Wang et al. [Reference 41, Line 64, Architecture and antigenicity of the Nipah virus attachment glycoprotein. Science, 2022. 375(6587): p.1373-1378.]

2. Figure 4A/Line 137: In line 137 Asn67 has been mentioned, but in panel A Asn77 has been shown.

Response: This is a typo, thank you for catching it. The correct number is Asn67, we corrected it in Figure 4A.

3. Line 148: Since for all the other mAbs the number of H-bond has been described, it would be appreciated if the number can be addressed for 1A9 as well.

Response: We amended the results section to clarify that the 1A9 interface features 12 hydrogen bonds (Line 162).

4. Line 223-224: Please cite the corresponding figure properly.

Response: The data referenced on Line 223-224 (n 3ow Lines are from previous papers, not this one. Both regarding neutralization of sera from NiV preF or postF immunized mice is shown in the two papers referenced.

5. Figure 1C: Please indicate what “*” means in the legend.

Response: The asterisks indicate antibodies that are neutralizing. We have modified the figure legends (Fig 1C and Supplementary Figure 1E) to include this information.

Fig 1C legend (Lines 314-316): “(C) Representative 2D class averages from negative stain EM of prefusion NiV F, alone and complexed with Fabs. Asterisks indicate antibodies that were shown to be neutralizing in the pseudovirus neutralization assay (Methods Section).”

Supplementary Fig 1E legend (Lines 365-367): “(E) Biolayer interferometry specificity binding for expressed monoclonal antibodies. Asterisks indicate antibodies that were shown to be neutralizing via pseudovirus neutralization assay.”

6. Figure 1E: The thickness of the lines inside the matrix is not intuitive. Please explain the meaning of the thicker lines in the legend.

Response: Thank you for bringing this to our attention. The lines were supposed to be boxes indicating the self-antibody competition (i.e. 5B3 vs 5B3). We corrected this in the figure and noted it in the legend as follows:

Lines 360-361: “Thicker boxes along the diagonal indicate self-antibody competition controls.”

7. Figure 3A-C,F: Since the volume data has been shown, it would be better to hide the ribbon representation for F and Fab, as D&E. And mentioning how many Fab densities have been observed for each map in the legend will be easier for the reader.

Response: Thank you for spotting this inconsistency. The ribbons have been removed from those panels. We also added text to the Figure 3 legend

(Lines 379-380): “The stoichiometry of the NiV F complex with Fab 2D3 was 1:1 (Fab:NiV F trimer), while the other complexes were all 3:1 (Fab:NiV F trimer).”

UNSOLICITED CHANGES

While preparing our responses to the reviewer comments, we made minor additions and changes to address omissions and typos. These changes do not alter the scientific substance of the manuscript. We kept them tracked in the manuscript and reproduce them here for convenience:

Lines 446-452: We added a figure legend description for Supplementary Figure 1, Panels F and G, which we had inadvertently omitted from our original submission:

“(F, G) Gating strategy for isolation of antigen-specific memory B cells by fluorescence activated cell sorting (FACS). The splenocytes were gated for antigen-specific memory B cells, specifically live IgG+ B cells which were also positive for the NiV prefusion F-APC probe (live, B220+, CD4-/CD8-/F4/80-/Gr-1-, IgM-, IgG+ and NiV prefusion F+). Cells index sorted 8 96-well plates using a FACS Aria II (BD Biosciences) interfaced with FacsDiva software (BD Biosciences). Flow cytometry analyses was performed using FlowJo software (Tree Star, Inc.).”

Lines 29-31: We added a passage to the abstract to reflect the results of the binding experiments suggested by reviewers:

“Antibodies bind the related Hendra virus (HeV) F protein, while sequence alignment of F proteins within the *Henipavirus* genus suggests that some of these newly identified neutralizing antibodies may also bind other *Henipavirus* F proteins.”

Reviewers' Comments:

Reviewer #1:

Remarks to the Author:

The authors have made a good faith effort to address the minor criticism of what I think was already a very strong paper. The additional analysis and experimental details added improve the clarity of the paper. This is an important contribution to the field of henipavirus vaccine development and monoclonal antibody therapeutics.

Reviewer #2:

Remarks to the Author:

Thanks for the revision. The authors addressed satisfactorily all the points reviewers raised, strengthening the manuscript. But minor clarification might still be needed:

Line 136-137: Correlated with the discussion section, it might be more clear if "binding affinity" can be specified as affinity of monovalent mAb-antigen interaction or binding affinity of Fab to exclude the avidity effect.

Line 139 "Initial screening of antibodies complexed.....": Later sentence mentioned method to prevent Fab aggregation. Should it also be Fab here?

Line 250: Seems like missing one ")"

Line 960: Figure 5 panels are mislabeled. And (A) looks like "top view" instead of "side view".

To the reviewers: Thank you for your responses and thoughtful consideration of our manuscript.

REVIEWERS' COMMENTS

Reviewer #1 (Remarks to the Author):

The authors have made a good faith effort to address the minor criticism of what I think was already a very strong paper. The additional analysis and experimental details added improve the clarity of the paper. This is an important contribution to the field of henipavirus vaccine development and monoclonal antibody therapeutics.

RESPONSE: Thank you.

Reviewer #2 (Remarks to the Author):

Thanks for the revision. The authors addressed satisfactorily all the points reviewers raised, strengthening the manuscript.

RESPONSE: Thank you.

But minor clarification might still be needed:

Line 136-137: Correlated with the discussion section, it might be more clear if "binding affinity" can be specified as affinity of monovalent mAb-antigen interaction or binding affinity of Fab to exclude the avidity effect.

RESPONSE: We made the clarification in the results section, indicating that Fabs were used.

Line 139 "Initial screening of antibodies complexed.....": Later sentence mentioned method to prevent Fab aggregation. Should it also be Fab here?

RESPONSE: We made the clarification in the text.

Line 250: Seems like missing one ")"

RESPONSE: We corrected this omission.

Line 960: Figure 5 panels are mislabeled. And (A) looks like "top view" instead of "side view".

RESPONSE: The figure panels are now labeled correctly and changes reflected in the figure legend.